# The natural pyrazolotriazine pseudoiodinine from *Pseudomonas mosselii* 923 inhibits plant bacterial and fungal pathogens

Ruihuan Yang [1,3], Qing Shi [2,3], Tingting Huang [2], Yichao Yan [1], Shengzhang Li [1], Yuan Fang [1], Ying Li [1], Linlin Liu [1], Longyu Liu [1], Xiaozheng Wang [2], Yongzheng Peng [1], Jiangbo Fan [1], Lifang Zou [1,2] ✉, Shuangjun Lin [2] & Gongyou Chen [1,2] ✉

Natural products largely produced by Pseudomonads-like soil-dwelling microorganisms are a consistent source of antimicrobial metabolites and pesticides. Herein we report the isolation of *Pseudomonas mosselii* strain 923 from rice rhizosphere soils of paddy fields, which specifically inhibit the growth of plant bacterial pathogens *Xanthomonas* species and the fungal pathogen *Magnaporthe oryzae*. The antimicrobial compound is purified and identified as pseudoiodinine using high-resolution mass spectra, nuclear magnetic resonance and single-crystal X-ray diffraction. Genome-wide random mutagenesis, transcriptome analysis and biochemical assays define the pseudoiodinine biosynthetic cluster as *psdABCDEFG*. Pseudoiodinine biosynthesis is proposed to initiate from guanosine triphosphate and 1,6-didesmethyltoxoflavin is a biosynthetic intermediate. Transposon mutagenesis indicate that GacA is the global regulator. Furthermore, two noncoding small RNAs, *rsmY* and *rsmZ*, positively regulate pseudoiodinine transcription, and the carbon storage regulators CsrA2 and CsrA3, which negatively regulate the expression of *psdA*. A 22.4-fold increase in pseudoiodinine production is achieved by optimizing the media used for fermentation, overexpressing the biosynthetic operon, and removing the CsrA binding sites. Both of the strain 923 and purified pseudoiodinine *in planta* inhibit the pathogens without affecting the rice host, suggesting that pseudoiodinine can be used to control plant diseases.

Rice (*Oryza sativa* L.) is a staple crop of global importance and is regarded as a strategic crop for food security by the Food and Agriculture Organization (FAO)[1]; unfortunately, rice production is hampered by multiple constraints including devastating diseases, like bacterial leaf blight (BLB), bacterial leaf streak (BLS), and rice blast which are incited by *Xanthomonas oryzae* pv. *oryzae* (*Xoo*), *X. oryzae* pv. *oryzicola* (*Xoc*), and *Magnaporthe oryzae*, respectively[2,3]. Currently, The cultivation of rice varieties with disease resistance (*R*) genes appears to be better option to control *X. oryzae* than other management schemes[2,4]; however, cultivars planted in China are commonly susceptible to the pathogens[5]. Although chemical fungicides and bactericides are frequently used to control rice diseases[6], their use has

[1]Shanghai Collaborative Innovation Center of Agri-Seeds/School of Agriculture and Biology, Shanghai Jiao Tong University, Shanghai 200240, China. [2]State Key Laboratory of Microbial Metabolism, School of Life Sciences & Biotechnology, Shanghai Jiao Tong University, Shanghai 200240, China. [3]These authors contributed equally: Ruihuan Yang, Qing Shi. ✉e-mail: zoulifang202018@sjtu.edu.cn; gyouchen@sjtu.edu.cn

resulted in pollution, drug-resistance and pathogen resurgence[7,8]. An environment-friendly control approach is the potential use of antagonistic bacteria as biocontrol agents (BCAs) that can suppress pathogens by producing bioactive secondary metabolites including antibiotics, siderophores and volatile compounds[9].

Natural products (NPs) are a consistent source of antimicrobial metabolites and drug leads and are largely produced by soil-dwelling microorganisms[10,11]. *Pseudomonas* species are gram-negative bacteria that persist in soil, water, animals, and the plant rhizosphere. Pseudomonads produce many antimicrobial NPs, including phenazine, pyrrole derivatives, 2,4-diacetylphloroglucinol (2,4-DAPG) and growth-promoting substances, which make them well adapted to environmental stress and suitable as BCAs of plant pathogens[12]. Many secondary metabolites in *Pseudomonas* are regulated by the GacS/GacA two-component system (TCS)[13], noncoding small RNAs (sRNAs) and CsrA/RsmA proteins[14]. With the advent of genomic sequencing, numerous antimicrobials have been discovered in the global microbiome[15] with benefits to modern agriculture[16] and human health[17]. However, the ongoing development of multidrug-resistant pathogens[18] has increased the urgency for discovering new NPs for controlling crop bacterial and fungal diseases.

Members of the naturally occurring heterocycle pyrazolo[4,3-*e*][1,2,4]triazine family have been isolated and characterized over the past 40 years[19], including fluviols[20], nostocine A[21] and pseudoiodinine[22]. The pseudoiodinine structure contains a 1,2,4-triazine moiety fused with a pyrazole ring and two methyl groups. Derivatives of the pyrazolotriazine family display a wide range of biological functions including antitumor, antiviral and antibacterial activities[23,24]. In particular, pseudoiodinine exhibited strong antineopleutic activities against sarcoma[25]. Interestingly, the biosynthetic route and biosynthetic gene clusters (BGCs) of pseudoiodinine have not been elucidated and remains a rich resource for synthetic biology and the development of bioactive derivatives.

In this work, *P. mosselii* strain 923 is screened for inhibition of *X. oryzae* and *M. oryzae*, and the antagonistic compound is identified as pseudoiodinine. The biosynthetic operon responsible for pseudoiodinine assembly is identified. Guanosine triphosphate (GTP) is shown to be a precursor and 1,6-didesmethyltoxoflavin (1,6-DDMT) is a biosynthetic intermediate of pseudoiodinine. Regulation of pseudoiodinine is shown to be mediated by GacA, *rsmY/Z*, and CsrA123, leading to genetically generate a pseudoiodinine-overproducing strain. The results show that pseudoiodinine is effective for controlling bacterial and fungus diseases of rice, thus expanding the range of potential options for controlling rice pathogens.

## Results

### Bioassay-directed screening afford strain 923 inhibiting rice pathogens

It is well-established that plants recruit beneficial bacteria to suppress pathogens in the rhizosphere[26]. Thus we investigated whether soil bacteria can suppress the growth of rice pathogens *Xoo* PXO99[A], *Xoc* RS105, and *M. oryzae* R01-1. A total of 223 cultivable bacterial isolates were obtained from 248 rhizosphere samples that were collected from 23 provinces in China; these were examined for antagonistic activity in vitro. A bacterial isolate named 923 showed strong inhibitory activity against *Xoo* PXO99[A], *Xoc* RS105, and *M. oryzae* R01-1 (Fig. 1a, b), and inhibition was generally greater for strains of *Xoo* than *Xoc* (Supplementary Fig. 1a, b). In addition, the strain 923 showed varying levels of antagonism towards 11 other *Xanthomonas* pathogens (Supplementary Fig. 1c).

To determine whether strain 923 inhibited the growth of *Xoo* and *Xoc in planta*, biocontrol experiments were conducted in the greenhouse. When either strain 923 or the supernatant (SUP) was applied as a pre-treatment (Pre), lesion lengths were about 50% smaller than the

control (Fig. 1c), which was consistent with the quantitative GUS assay results (Fig. 1d). Moreover, both pre- and post-treatment with the strain 923 resulted in significantly smaller lesions, caused by *Xoo* and *Xoc*, respectively, on rice leaves in the field (Fig. 1e, f). These results suggest that the strain 923 is an effective biocontrol agent (BCA) for bacterial pathogens.

### Taxonomic characterization of strain 923 as *P. mosselii*

Multiple approaches were used to identify the strain 923. Phylogenetic analysis of *16S rRNA* sequences showed that 923 was a member of the genus of *Pseudomonas* and grouped with *P. mosselii* DSM17497, which was 99.6% similar (Supplementary Fig. 2b). To further confirm the taxonomic classification of the strain 923, the whole genome was sequenced (Supplementary Fig. 2a). Genomic-based average nucleotide identity (ANI) analysis indicated that the 923 genome had an ANI value of 99.25% when compared with *P. mosselii* DSM17497; this is higher than the threshold value of 95% for species demarcation, confirming that the strain 923 is *P. mosselii*. Additional phylogenomic assays further supported these conclusions (Supplementary Fig. 2c).

### Pseudoiodinine is the antimicrobial compound produced by *P. mosselii* 923

Antibiotic-producing bacteria generally produce inhibition zones when they are spotted onto media overlaid with a target pathogen[27]. We prepared extracts from the strain 923 cultures with a variety of organic solvents and spotted these to plates seeded with *Xoo* PXO99[A]. Ethyl acetate extracts produced the largest inhibition zones when spotted to the *Xoo* PXO99[A] overlay (Supplementary Fig. 3a). Furthermore, the ethyl acetate extract was subjected to C18 reversed-phase silica gel column to yield twelve fractions, labeled as A1-A12. Based on the bioactivity profile, fractions A1 and A5-A6 were further purified by preparative HPLC on a C18 column to give compounds PM-1 to PM-7 (Fig. 2a and Supplementary Fig. 3a). Among the seven purified compounds, only PM-3 displayed significant inhibitory activity for *Xoo* PXO99[A] (Fig. 2b). The molecular formula of PM-3 was determined to be $C_6H_7N_5O$ by high-resolution mass spectra (HRMS) (m/z 166.0722 $[M+H]^+$, calculated for $C_6H_8N_5O$, 166.0729) (Supplementary Fig. 4). The nuclear magnetic resonance (NMR) data (Supplementary Figs. 5–8) and single-crystal X-ray diffraction (Fig. 2c and Supplementary Table 1) further confirmed the structure of PM-3, which was identical to pseudoiodinine. Furthermore, PM-3 (pseudoiodinine) demonstrated efficacy against many other pathogenic *Xanthomonas* spp., including *X. campestris* pv. *phaseoli*, *X. campestris* pv. *malvacearum* and *X. citri* subsp. *citri* (Supplementary Fig. 3d).

### Antimicrobial activities of Pseudoiodinine

MIC (minimum inhibitory concentration) and $EC_{50}$ (effective concentration for 50% inhibition) assays showed that pseudoiodinine was more toxic to *Xoo* PXO99[A] and *Xoc* RS105 than other 11 *Xanthomonas* spp. with MICs of 0.5 and 4 µg/mL and $EC_{50}$ values of 0.17 and 1.36 µg/mL, respectively (Supplementary Fig. 9a, b and Table 1). Pseudoiodinine also inhibited the growth of *M. oryzae* R01-1 with MIC and $EC_{50}$ values of 8.25 and 4.43 µg/mL, respectively (Supplementary Fig. 9c).

Prior studies demonstrated that phenazine-1-carboxylic acid (PCA; also known as Shenqinmycin) was an effective bactericide for control of BLB and BLS in China[28,29]. Surprisingly, our results show that the MIC and $EC_{50}$ values of pseudoiodinine were significantly lower than PCA (Table 1), further supporting the potential of pseudoiodinine as an eco-friendly biopesticide.

*P. mosselii* 923 and pseudoiodinine were examined for their ability to inhibit pathogens *Xoc* and *Xoo* and were shown to significantly reduce the lesion area of rice leaves at 15 dpi (Supplementary Fig. 10a–c). Meanwhile, measurement of the bacterial population in rice leaves indicated that the Tre (treatment) or Pre treatments significantly reduced colonization by *Xoo* and *Xoc* at 14 and 21 dpi

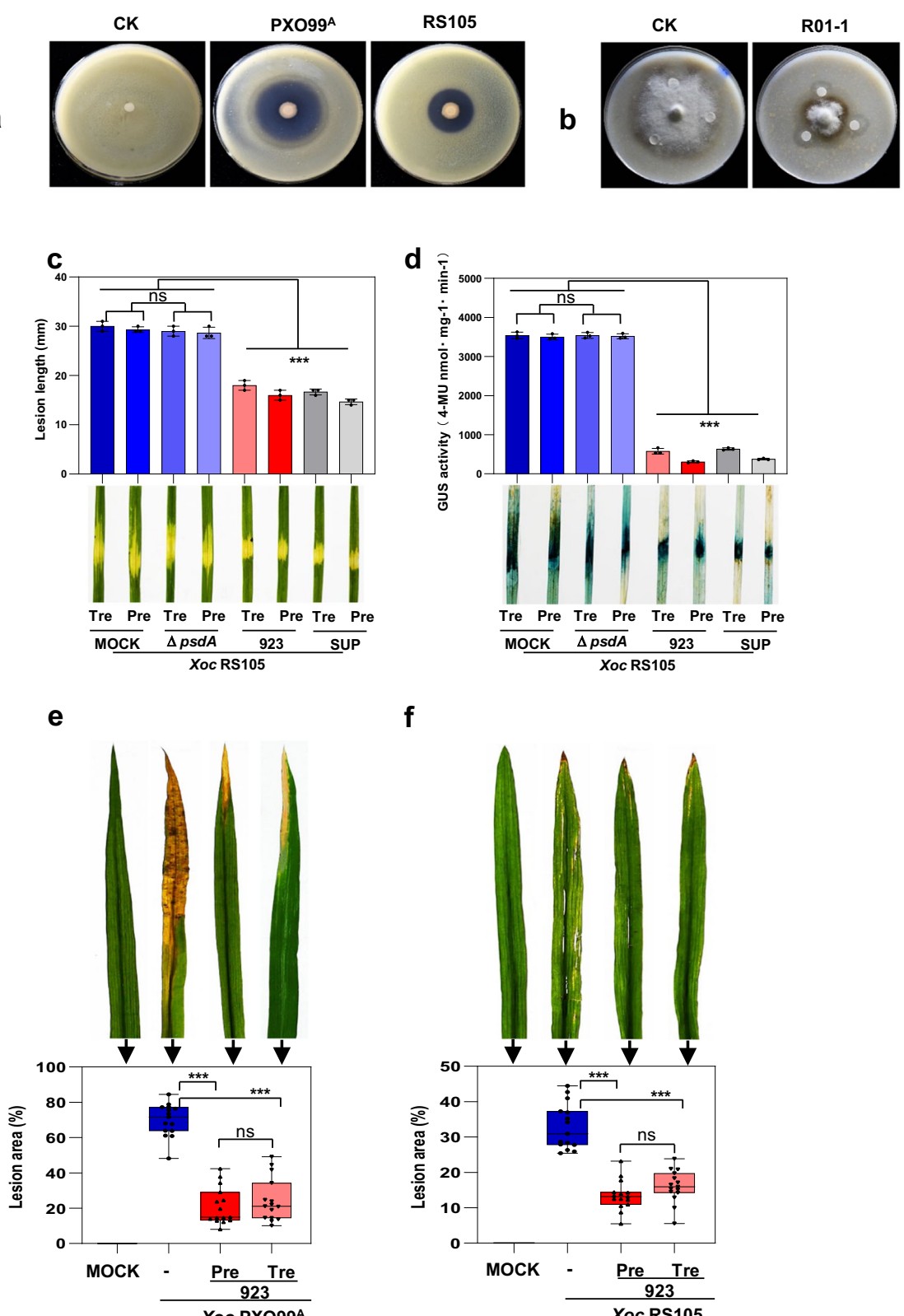

(Supplementary Fig. 10b–d). In order to identify whether pseudoiodinine is sufficient to protect rice leaves from lesions, we retested the biocontrol activity of mutant Δ*psdA* and WT of *P. mosselii* 923 on rice seedlings and adult plants. Leaves inoculated with the deletion mutant Δ*psdA* exhibited lesion lengths and the GUS activity equivalent to the MOCK treatment (Fig. 1c, d). Meanwhile, we tested the effects of mutant Δ*psdA*, WT *P. mosselii* 923 and pseudoiodinine on the growth

inhibition of *Xoo in planta*. As predicted, the deletion mutant Δ*psdA* induced lesions that were similar in size to the MOCK treatment (Supplementary Fig. 11), implying the necessity only for pseudoiodinine contributes to plant protection. The efficacy of pseudoiodinine was further analyzed by applying concentrations from 0 to 20 μM and evaluating disease in field-grown rice inoculated with *Xoo* PXO99[A] as a pre- or post-treatment. Pseudoiodinine at concentrations >5 μM

**Fig. 1 | Antagonistic and biocontrol activity of *P. mosselii* 923. a, b** Antagonistic activity of strain 923 towards *Xoo* PXO99[A], *Xoc* RS105, and *M. oryzae* isolate R01-1 in co-cultured assays, untreated strain 923 was used as control (CK). **c** *Xoc* RS105-Gus lesion lengths on Yuanfengzao rice inoculated with strain 923, Δ*psdA* or 923 SUP at 7 d post-inoculation (dpi) in the greenhouse. 'Pre' represents rice leaves pretreated with water (MOCK-Pre), 923, Δ*psdA* or 923 SUP (supernatant) prior to pathogen inoculation; 'Tre' represents rice leaves inoculated with *Xoc* RS105-Gus and then treated with water (MOCK-Tre), 923, Δ*psdA* or 923 SUP. **d** Population dynamics of *Xoc* RS105-Gus in rice leaves. Bacterial populations were monitored by the GUS quantification and histochemical staining at 7 dpi. Error bars show means ± SD ($n = 3$ independent leaves) and significant differences at ***$P < 0.001$,

***$P = 6.4 \times 10^{-14}$, $= 3.2 \times 10^{-15}$, $= 8.3 \times 10^{-15}$, $= 5.5 \times 10^{-16}$ in sequence of **c**, ***$P < 0.001$, ***$P = 1.3 \times 10^{-25}$, $= 2.4 \times 10^{-26}$, $= 1.8 \times 10^{-25}$, $= 3.6 \times 10^{-26}$ in sequence of **d**, ns not significant ($P > 0.05$). **e, f** Disease lesions areas on field-grown rice inoculated with strain 923 and (**e**) *Xoo* PXO99[A] or (**f**) *Xoc* RS105. Lesion areas were determined at 15 dpi ($n = 15$ independent leaves, means ± SD), ***$P < 0.001$, ***$P = 1.2 \times 10^{-15}$, $= 1.3 \times 10^{-14}$ in sequence of **e**; ***$P < 0.001$, ***$P = 4.7 \times 10^{-13}$, $= 3.2 \times 10^{-11}$ in sequence of **f**; ns, not significant ($P > 0.05$). The statistical significance of the lesion areas inoculated with *Xoc* or *Xoo* was determined using the LSD test method and one-way ANOVA; The center bar represents the mean, and Min to Max of box and whiskers was used. The experiments were repeated three times independently with similar results.

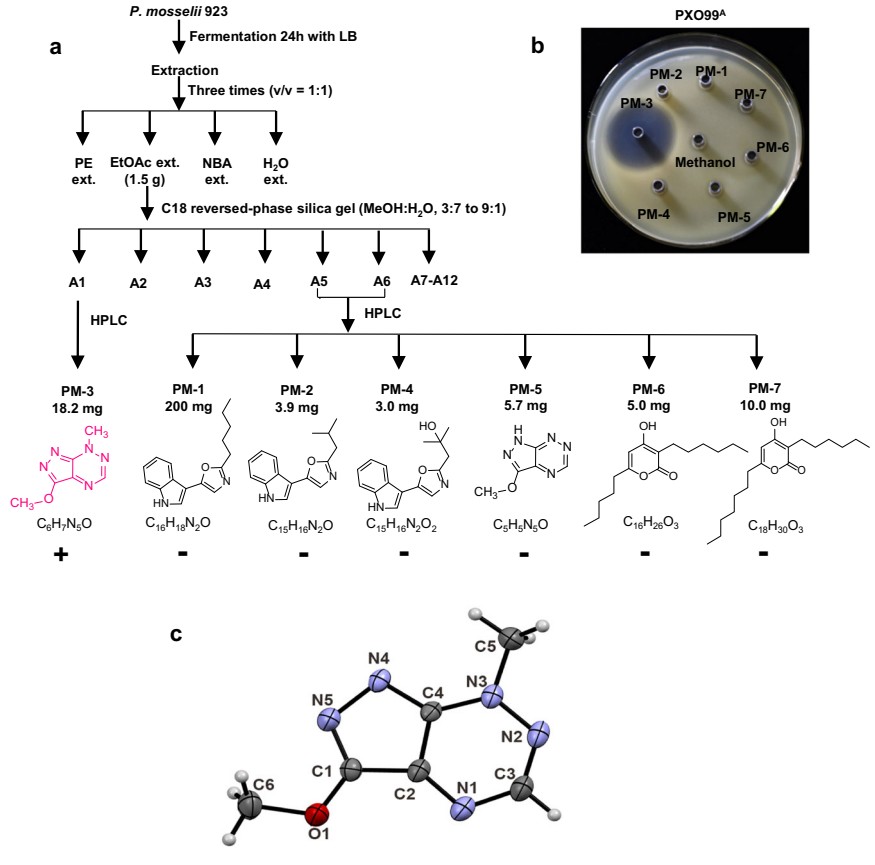

**Fig. 2 | Extraction and purification of PM-3 (pseudoiodinine). a** Flow chart for extraction and isolation from the strain 923 compounds that inhibited PXO99[A]. Abbreviations and structures are as follows: PE, petroleum ether; EtOAc, ethyl acetate; NBA, n-butyl alcohol; MeOH, methanol; compounds: PM-1, $C_{16}H_{18}N_2O$; PM-2,

$C_{15}H_{16}N_2O$; PM-3, $C_6H_7N_5O$; PM-4, $C_{15}H_{16}N_2O_2$; PM-5, $C_5H_5N_5O$; PM-6, $C_{16}H_{26}O_3$; and PM-7, $C_{18}H_{30}O_3$. **b** The antibacterial activity of compounds PM-1 to PM-7 for *Xoo* PXO99[A]; methanol was used as a negative control. **c** X-ray ORTEP drawing for PM-3.

caused a significant reduction in lesion size (Supplementary Fig. 3b, c), indicating that pseudoiodinine effectively inhibits the growth of *Xoo* PXO99[A] and *Xoc* RS105 in rice tissue. Interestingly, we observed that pseudoiodinine also significantly reduced lesion area of rice blast caused by the fungus *M. oryzae* R01-1 (Supplementary Fig. 12). The above data demonstrate that the compound is a promising biopesticide both for bacterial and fungal diseases in rice.

### Identification of genes for pseudoiodinine biosynthesis

Although pseudoiodinine was first identified in *P. fluorescens* var. *pseudoiodinine*[22], the genes involved in its synthesis were not characterized. The genome of *P. mosselii* 923 was analyzed for secondary metabolites using the antiSMASH program[30] (https://antismash.secondarymetabolites.org/). However, it was not certain which cluster was associated with pseudoiodinine biosynthesis.

To identify genes in the pseudoiodinine pathway, the EZ-Tn5 transposome system was used to generate random mutations in *P. mosselii* 923. Approximately 10,000 mutants of the strain 923 were screened for antimicrobial activity towards *Xoo* PXO99[A], and mutant 34-6 was devoid of antagonistic activity in vitro (Fig. 3a). The location of the single copy Tn5 insertion in mutant 34-6 mapped to *gacA* in the genome (Fig. 3a); interestingly, *gacA* encodes a response regulator that is highly conserved in *Pseudomonas* spp. and belongs to the two-component regulatory system (TCS) family[13]. To further confirm *gacA* function in pseudoiodinine synthesis, we generated a markerless knockout mutant where the complete *gacA* was deleted from *P. mosselii* 923 by double homologous recombination. Pseudoiodinine production was abolished in the resulting mutant, which was designated Δ*gacA* (Supplementary Fig. 13b, c). Complementation of the Δ*gacA* mutant with gacA in trans restored antibacterial activity and

**Table 1 | MIC and EC$_{50}$ values of pseudoiodinine and phenazine-1-carboxylic acid (PCA)**

| Pathogens | Concentration (µg/mL) | | | |
|---|---|---|---|---|
| | Pseudoiodinine | | PCA | |
| | MIC | EC$_{50}$ | MIC | EC$_{50}$ |
| *X. oryzae* pv. *oryzae* PXO99$^A$ | 0.5 | 0.17 ± 0.02 | 1 | 0.26 ± 0.05 |
| *X. oryzae* pv. *oryzicola* RS105 | 4 | 1.36 ± 0.26 | 8 | 4.01 ± 0.58 |
| *X. translucens* pv. *cerealis* NXtc01 | 5 | 2.19 ± 0.44 | 16 | 7.75 ± 1.44 |
| *X. citri* subsp. *citri* 029-1 | 5 | 1.76 ± 0.13 | 128 | 38.41 ± 4.50 |
| *X. campestris* pv. *phaseoli* ICMP5834 | 5 | 1.51 ± 0.16 | 128 | 39.34 ± 1.51 |
| *X. campestris* pv. *vesicatoria* NCPPB701 | 8 | 2.93 ± 0.34 | 128 | 35.67 ± 2.64 |
| *X. axonopodis* pv. *vignicola* ATCC11648 | 8 | 2.71 ± 0.59 | >128 | 55.30 ± 2.90 |
| *X. axonopodis* pv. *glycines* ICMP5732 | 8 | 2.41 ± 0.30 | >128 | 51.40 ± 3.39 |
| *X. axonopodis* pv. *allii* LMG578 | 10 | 3.64 ± 0.23 | >128 | 64.92 ± 4.55 |
| *X. campestris* pv. *musacearum* ICMP287 | 10 | 4.00 ± 0.51 | 128 | 39.01 ± 2.83 |
| *X. campestris* pv. *malvacearum* ATCC12131 | 16 | 6.65 ± 0.43 | >128 | 51.60 ± 1.96 |
| *X. arboricola* pv. *juglandis* DW3F3 | 16 | 5.38 ± 0.48 | 128 | 42.42 ± 0.42 |
| *X. campestris* pv. *campestris* 8004-WT | 48 | 22.06 ± 2.55 | 128 | 43.82 ± 1.42 |

Note: The EC$_{50}$ was presented as the mean ± SD (*n* = 3).

pseudoiodinine production to the wild-type level, but the over-production of gacA in trans did not increase pseudoiodinine production in the strain 923 (Supplementary Fig. 13a–c). These results suggest that GacA regulates pseudoiodinine biosynthesis, being consistent with GacA role in the regulation of other secondary metabolites in *Pseudomonas*[13,31,32].

To further identify the biosynthetic gene cluster responsible for pseudoiodinine production, the transcriptomes of *P. mosselii* 923 (WT) and the Δ*gacA* knockout (KO) mutant were compared. This result revealed that 604 genes were downregulated in the *gacA* mutant (Supplementary Fig. 13d, e); and 29 differentially expressed genes (DEGs) were highly upregulated in the WT as compared to the KO mutant (Supplementary Fig. 14 and Supplementary Table 2). To verify the RNA-seq data, fifteen DEGs were randomly selected for secondary confirmation by RT-qPCR analysis (Supplementary Fig. 15). Nineteen genes were cloned in the suicide vector p2P24Km and are listed in Supplementary data 1 as p24-*orf2059* – p24-*orf2077*. Deletion of these 19 genes resulted in *P. mosselii* mutants Δ*orf2059*–Δ*orf2077* (Supplementary Data 1). The 19 mutants were screened for anti-microbial activity and pseudoiodinine biosynthesis, and seven mutants (Δ*orf2064*–Δ*orf2070*) were impaired in production of pseudoiodinine and defective in antimicrobial activity. These seven mutants were renamed as follows: Δ*psdA* (Δ*orf2064*); Δ*psdB* (Δ*orf2065*); Δ*psdC*, (Δ*orf2066*); Δ*psdD* (Δ*orf2067*); Δ*psdE* (Δ*orf2068*); Δ*psdF* (Δ*orf2069*); and Δ*psdG* (Δ*orf2070*). The seven genes encoding pseudoiodinine, namely *psdA, psdB, psdC, psdD, psdE, psdF,* and *psdG*, were cloned in plasmid pBSPPc downstream of the *psdA* promoter, resulting in pBS-*psdA*, pBS-*psdB*, pBS-*psdC*, pBS-*psdD*, pBS-*psdE*, pBS-*psdF* and pBS-*psdG*, respectively. These clones were transformed into the seven corresponding mutants to obtain the corresponding complemented strains (CΔ*psdA*-CΔ*psdG*), which were rescued for antibacterial activity and pseudoiodinine production (Fig. 3b–d).

RT-PCR confirmed that *psdABCDEFG* in the *psd* gene cluster originated from a single operon (Fig. 3e and Supplementary Fig. 13f). This cluster (*psdABCDEFG*) was validated by heterologous expression in *P. putida* KT2440. The *psdABCDEFG* under native promoter endows

*P. putida* KT2440 with the capacity to produce pseudoiodinine, showing that the seven genes are sufficient (Fig. 4a).

Further homology analysis results showed that the *psdABCDEFG* gene cluster was conserved in other pseudomonads; especially in the type strain *P. mosselii* DSM17497, which has a 99% amino acid similarity (Supplementary Fig. 16). However, *P. mosselii* DSM17497 did not produce detectable levels of pseudoiodinine and did not inhibit *Xoo* PXO99$^A$ (Supplementary Fig. 17a, b). Complementing the DSM17497 *psd* operon (*orf2184-orf2190*) in the *P. mosselii* 923 mutants Δ*psdB*, Δ*psdC* and Δ*psdG*, respectively, the bacteriostatic activity and pseudoiodinine production can be restored to the wild-type 923 level (Supplementary Fig. 17a, b). Moreover, the seven genes of *orf2184-orf2190* were expressed in *P. mosselii* DSM17497 (Supplementary Fig. 17c), indicating that the pseudoiodinine gene cluster was transcribed in *P. mosselii* DSM17497. Above results indicated that the *psdABCDEFG* gene cluster was functional in the *P. mosselii* DSM17497, but the lack of pseudoiodinine production warrants further study.

**Proposed biosynthetic pathway for pseudoiodinine**

BLAST analysis showed that the amino acid sequences of PsdD and PsdG have significant similarity to GTP cyclohydrolase II and deaminase, respectively, which are involved in riboflavin and toxoflavin synthesis[33,34]. Considering the correlation between the structures of pseudoiodinine and 1,6-didesmethyltoxoflavin (1,6-DDMT), we present a hypothetical biosynthetic pathway for pseudoiodinine based on similarity to toxoflavin biosynthesis[33] (Fig. 4b). In the proposed pathway, PsdD and PsdG use GTP to generate 2,5-diamino-6-(5-phospho-*d*-ribosylamino) pyrimidin-4(*3H*)-one (structure 1, Fig. 4b) and 5-amino-6-(5-phospho-*d*-ribitylamino) uracil (structure 2, Fig. 4b) in the first two steps. Then, either PsdE and/or PsdC mediate formation of the N−N bond formation by catalyzing the conjugation of structure 2 or the dephosphorylated product 5-amino-6-*d*-ribitylaminouracil (structure 3, Fig. 4b) with glycine to generate 1,6-DDMT. Finally, 1,6-DDMT would be subjected to a sequence of ring contraction and methylation steps to produce pseudoiodinine. Although 1,6-DDMT was not observed in either wild-type *P. mosselii* 923 or mutant strains, pseudoiodinine production was restored when 1,6-DDMT was fed to the mutant strains Δ*psdC* and Δ*psdE* (Supplementary Fig. 18), demonstrating that 1,6-DDMT is a key intermediate.

**The GacA-CsrA regulatory systems control pseudoiodinine synthesis**

The GacS/GacA TCS controls the expression of genes involved in the biosynthesis of multiple antimicrobial compounds and lytic enzymes[35]. In response to stimuli, GacS autophosphorylates and activates the response regulator GacA by phosphotransfer[31]. GacA activates the expression of the *rsmX, rsmY,* and *rsmZ* sRNAs by binding a conserved upstream activation sequence (UAS) TGTAAGNNATNNCTTACA[32] that regulates the expression of target genes.

In *P. mosselii* 923, two noncoding small RNAs, *rsmY* and *rsmZ*, contained the GacA-binding motif (5′TGTAAGCNAANGCTTACA3′) in their promoter regions (Supplementary Fig. 19b, c). To investigate the potential interaction of *rsmY* and *rsmZ* with GacA, the latter was overproduced in *E. coli* BL21 (DE3), purified, and shown to bind the *rsmY* and *rsmZ* promoter regions in electrophoretic mobility shift assays (EMSA) (Supplementary Fig. 19a−c). Interestingly, GacA did not interact with the *psdA* promoter (data not shown). Deletions in *rsmY* and *rsmZ* were constructed to further investigate their roles in pseudoiodinine production and activity. Individual Δ*rsmY* and Δ*rsmZ* mutants had no effect on antibacterial activity; however, the double mutant Δ*rsmYZ* was almost completely devoid of antibacterial activity against *Xoo* PXO99$^A$ (Fig. 5a). Antibacterial activity was restored to wild-type levels in the three complemented strains, namely CΔ*rsmY*, CΔ*rsmZ,* and CΔ*rsmYZ* (Fig. 5a). Pseudoiodinine production was

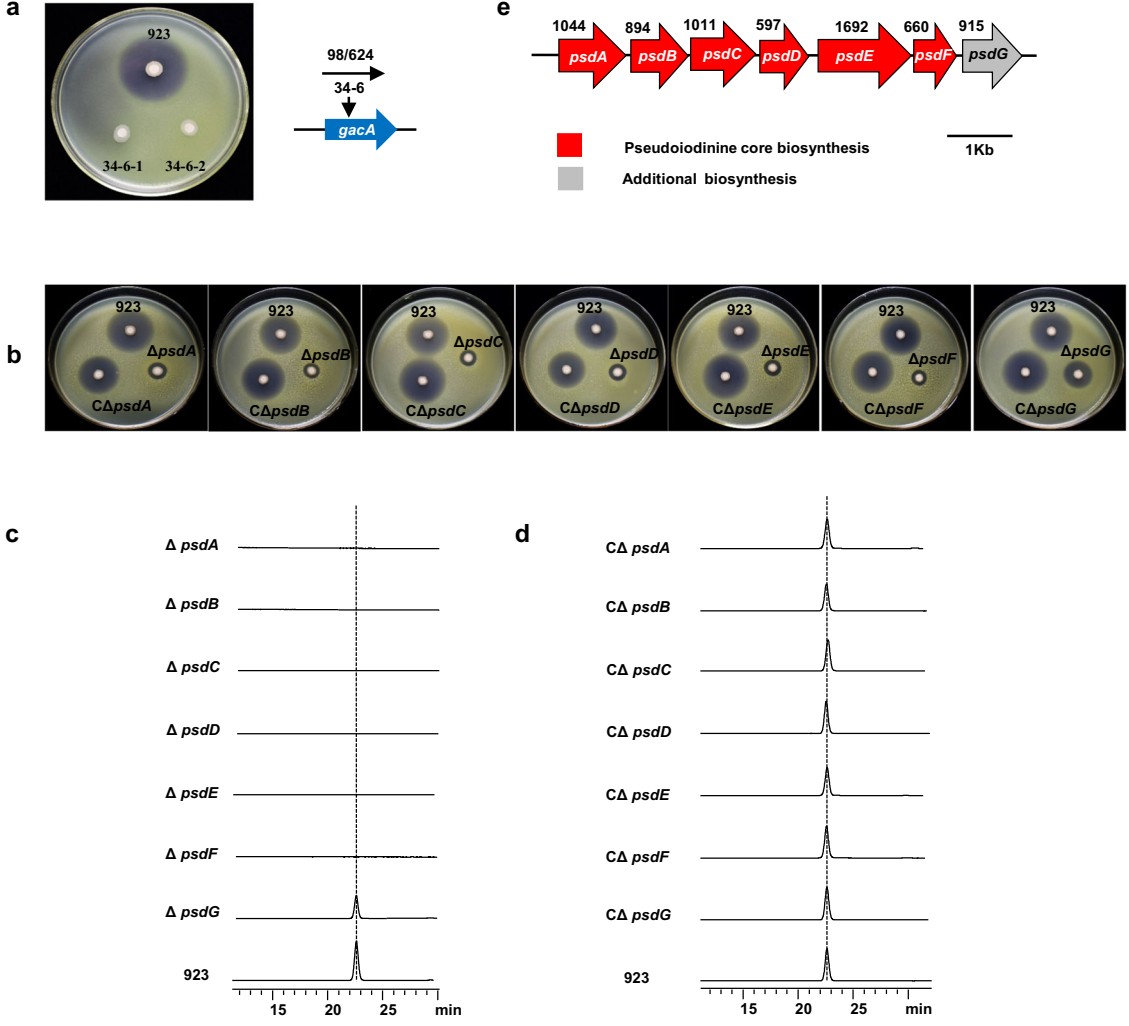

**Fig. 3 | Genetic determinants of pseudoiodinine biosynthesis. a** Mutant 34−6 of *P. mosselii* 923 lacks antimicrobial activity for *Xoo* PXO99[A] and contains a Tn5 insertion in *gacA*. **b** Antibacterial activity assays of *P. mosselii* 923, Δ*psdA*, Δ*psdB*, Δ*psdC*, Δ*psdD*, Δ*psdE*, Δ*psdF* and Δ*psdG* mutants and their corresponding complemented strains (CΔ*psdA* - CΔ*psdG*). **c** HPLC analysis (Method B, detected at 500 nm) of mutants Δ*psdA*, Δ*psdB*, Δ*psdC*, Δ*psdD*, Δ*psdE*, Δ*psdF*, and Δ*psdG*. **d** HPLC analysis (Method B, detected at 500 nm) of complemented mutants CΔ*psdA* - CΔ*psdG*. **e** Schematic of the pseudoiodinine biosynthetic gene cluster in *P. mosselii*

923. Numbers above the gene names indicate the ORF length. Colors denote gene functions as follows: red, primary biosynthetic genes; gray, additional biosynthetic genes. The seven gene annotations are displayed as follows: *psdA*, methyl-transferase type 12; *psdB*, glyoxylase bleomycin resistance protein dioxygenase superfamily protein; *psdC*, sulphatase-modifying factor protein; *psdD*, GTP cyclo-hydrolase; *psdE*, WD-40 repeat protein; *psdF*, methyltransferase domain; *psdG*, riboflavin biosynthesis protein RibD.

significantly reduced in the Δ*rsmY* mutant, but not in Δ*rsmZ* mutant. As predicted from the antibacterial assays, the Δ*rsmYZ* double mutant did not produce pseudoiodinine. The two complemented strains (CΔ*rsmY* and CΔ*rsmZ*) were partially rescued for pseudoiodinine production, whereas simultaneous complementation with *rsmY* and *rsmZ* restored pseudoiodinine production to wild-type levels (Fig. 5b, c). These data indicate that GacA positively regulated pseudoiodinine biosynthesis by directly binding to the *rsmY* and *rsmZ* promoters, which activates transcription of the operon in *P. mosselii* 923.

RsmA (repressor of secondary metabolites)[36] and its homolog CsrA (carbon storage regulator A)[37] are RNA-binding proteins that interact with the 5′ untranslated GGA motif of mRNAs near the ribosome-binding site (RBS) of target genes, thus blocking ribosome access[38]. CsrA/RsmA proteins can be sequestered by regulatory RNAs, which relieves translational repression of target mRNAs[39]. We investigated whether CsrA/RsmA contributed to pseudoiodinine synthesis in *P. mosselii* 923. BLAST algorithms were used to search for CsrA/RsmA orthologs in the *P. mosselii* 923 genome, and three homologous genes were identified, namely *csrA1*, *csrA2*, and *csrA3*. CsrA2 and CsrA3 showed

more than 75% amino acid similarity, whereas CsrA1 was less than 50% similar (Supplementary Fig. 19d).

Deletion mutagenesis and a *psdA*::*uidA* transcriptional fusion were used to investigate whether CsrA1, CsrA2, and CsrA3 were involved in *psdA* expression. We analyzed *csrA1*, *csrA2*, and *csrA3* expression in wild-type 923 and in the Δ*csrA1*, Δ*csrA2*, and Δ*csrA3* mutants by measuring activity of the *psdA* promoter. Quantitative GUS assays showed that *psdA* promoter-driven GUS activity was dramatically higher in the Δ*csrA1*, Δ*csrA2*, and Δ*csrA3* mutants than in the wild-type strain 923 (Supplementary Fig. 19f). These results indicated that the deletion of *csrA1*, *csrA2*, and *csrA3* increased expression of *psdA*, which suggests that the CsrA proteins repress transcription of the pseudoiodinine biosynthesis operon. To further investigate whether CsrA1, CsrA2, and CsrA3 were required for improving pseudoiodinine production, additional deletion mutants were constructed, namely Δ*csrA1A2*, Δ*csrA1A3*, Δ*csrA2A3*, and Δ*csrA1A2A3*. Pseudoiodinine production was significantly improved in the Δ*csrA2*, Δ*csrA3*, Δ*csrA1A2*, Δ*csrA1A3*, Δ*csrA2A3* and Δ*csrA1A2A3* mutants but not the Δ*csrA1* mutant strain (Fig. 5d, e), which was consistent with antibacterial assays against *Xoo*

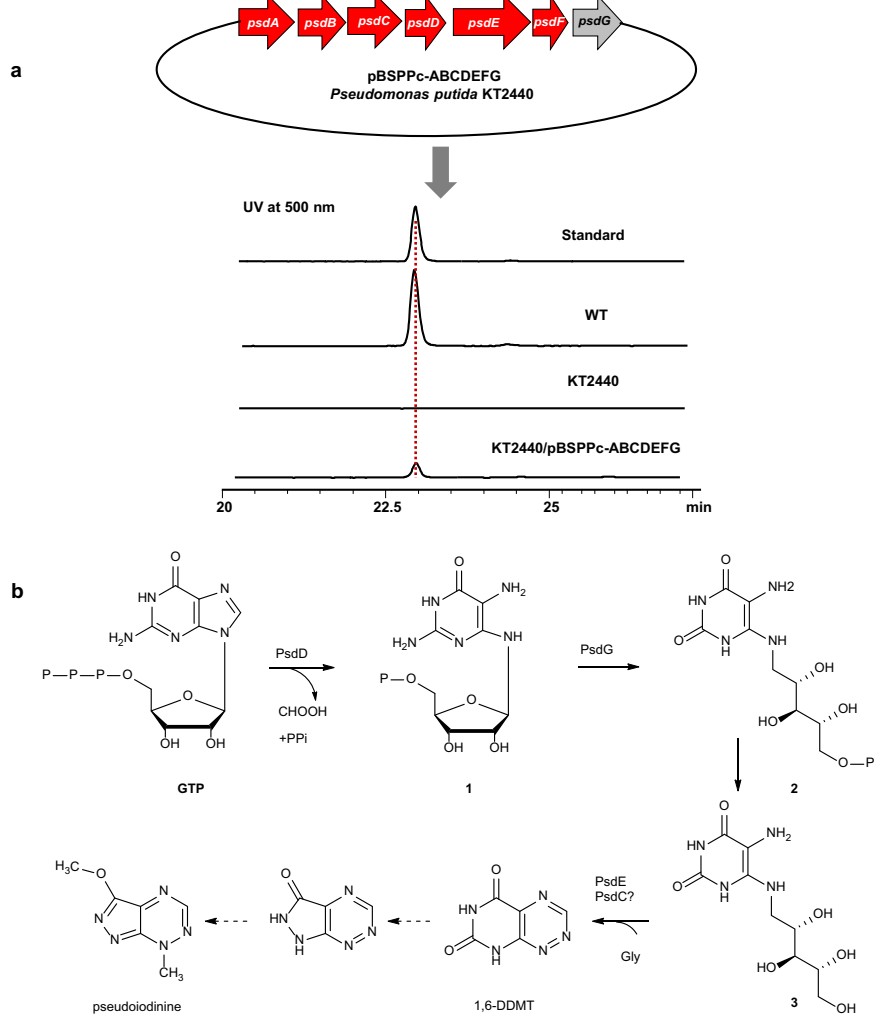

**Fig. 4 | Heterologous expression of the *psdABCDEFG* operon in *P. putida* KT2440 and proposed biosynthetic route for pseudoiodinine in *P. mosselii* 923.** **a** Pseudoiodinine biosynthetic cluster in construct pBSPPc-ABCDEFG and HPLC analysis (Method B, detected at 500 nm) of the fermentation extracts of WT, KT2440 and engineered heterologous expression strain and the standard sample (pseudoiodinine). Standard, pseudoiodinine isolated from *P. mosselii* 923; WT, wild-type *P. mosselii* 923; KT2440, *P. putida*; and KT2440/pBSPPc-ABCDEFG, *P. putida* containing the pseudoiodinine gene cluster in pBSPPc. **b** Proposed pathway for pseudoiodinine biosynthesis. **1**: 2,5-diamino-6-(5-phospho-*d*-ribosylamino) pyrimidin-4(*3H*)-one; **2**: 5-amino-6-(5-phospho-*d*-ribitylamino) uracil; **3**: 5-amino-6-*d*-ribitylaminouracil; 1,6-DDMT: 1,6-didesmethyltoxoflavin.

PXO99[A] (Supplementary Fig. 19e). These results indicate that CsrA1, CsrA2 and CsrA3 negatively impact *psdA* expression and pseudoiodinine production, and CsrA2 and CsrA3 function as major regulators (Fig. 5e, f).

### Engineering *P. mosselii* 923 to improve pseudoiodinine production

The *psdABCDEFG* operon was used to construct the genetically-modified strain *P. mosselii* Δ*csrA1A2A3*-pBSPPc-P2064-RBS-Pseu-ORF, which was optimized for higher pseudoiodinine production based on the GacA-*rsmY*/*Z*-CsrA123 regulatory system. Cultivation of *P. mosselii* 923 in Tryptic soy broth (TSB) resulted in increased growth and significantly higher levels of pseudoiodinine after a 36-h fermentation as compared to LB medium (Supplementary Fig. 20a, b); thus, TSB medium was chosen for further experiments. Next, *csrA1, csrA2*, and *csrA3* genes were deleted in strain 923, leading to 12.5 mg/L pseudoiodinine versus the 1.9 mg/L produced by the wild-type strain 923 (Fig. 5e). The *psdABCDEFG* operon was then introduced into the Δ*csrA1A2A3* mutant to obtain strain Δ*csrA1A2A3*-pBSPPc-ABCDEFG (Supplementary data 1), which resulted in a 16.6-fold increase in pseudoiodinine production relative to the 923 strain (Fig. 5e). Given that the RBS sequence of the

*uidA* gene[40] could be bound and negatively regulated by RsmA[14], we measured GUS activity at the transcriptional and post-transcriptional levels in the wild-type 923 and the Δ*csrA1A2A3* mutant. The quantitative assays showed that the GUS activity at the transcriptional level was dramatically higher than the post-transcriptional level (Supplementary Fig. 20c); this indicates that ribosome binding at *uidA* was higher than binding to *psdA*. Therefore, the RBS of the operon was edited to be negatively regulated by CsrA1, CsrA2, and CsrA3 by one-step cloning technology (Supplementary Fig. 20d). Finally, the yield of pseudoiodinine by the modified strain was increased to 42.5 mg/L, which was more 22.4-fold higher than the wild-type *P. mosselii* 923 (1.9 mg/L) (Fig. 5e). Meanwhile, we compared antibacterial activity and growth for the WT and the overproducing strains, and the results indicated that pseudoiodinine overproduction improved antibacterial activity against both *Xoo* PXO99[A] and *Xoc* RS105 pathogens (Supplementary Fig. 21a–c).; however, there was a decrease in the growth of *P. mosselii* in the initial 14 h of fermentation (Supplementary Fig. 21d–f).

### Discussion

Beneficial microorganisms are abundant in agroecosystems and function to control plant diseases and promote plant growth[26].

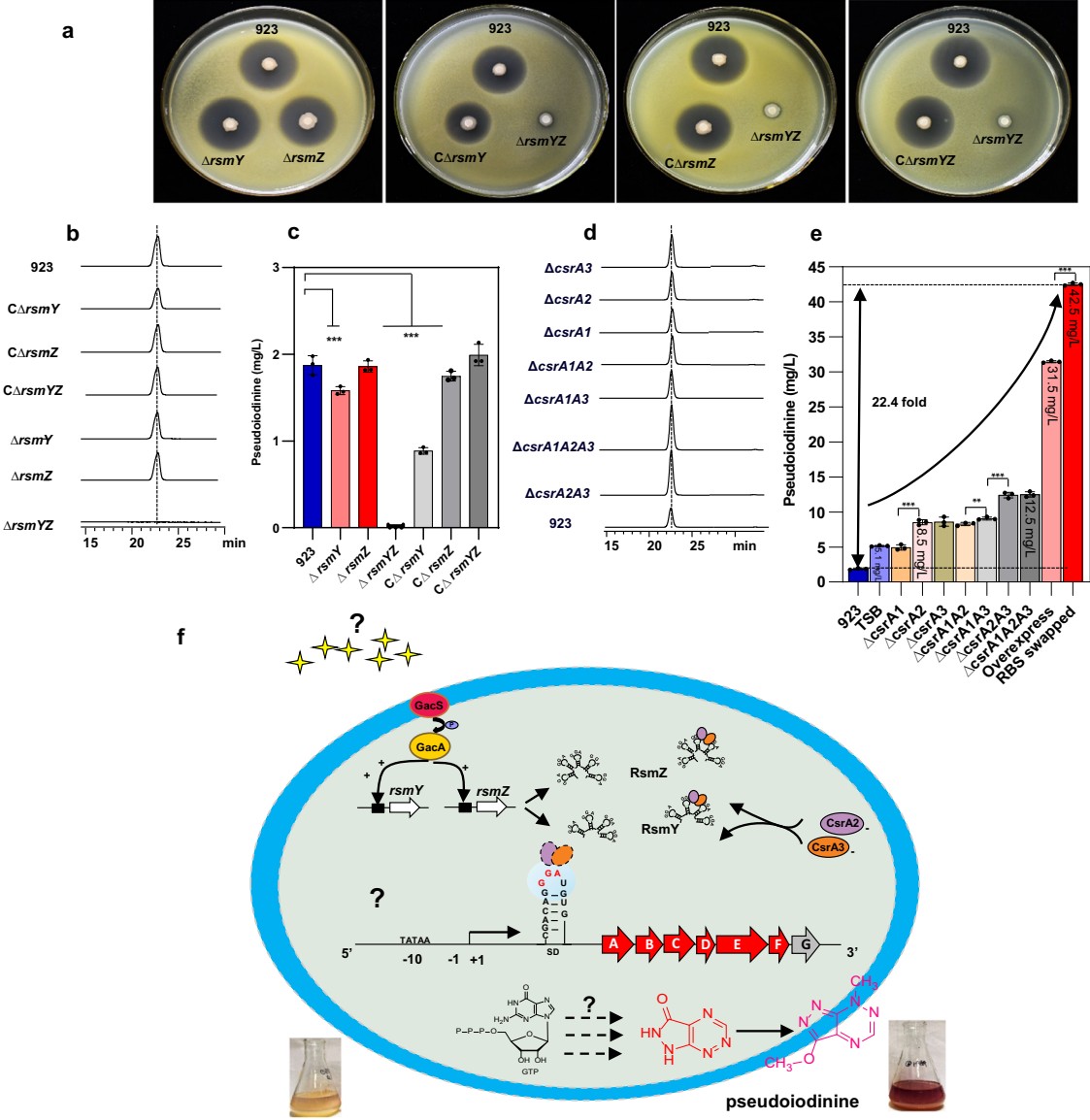

**Fig. 5 | GacA and CsrA mediate the biosynthesis of pseudoiodinine.**
**a** Antibacterial activity assays of *P. mosselii* 923, Δ*rsmY*, Δ*rsmZ*, and Δ*rsmYZ* mutant and complemented mutants CΔ*rsmY*, CΔ*rsmZ*, and CΔ*rsmYZ*. Panels **b**, **d**, HPLC profiles (Method B, detected at 500 nm) of the mutants and complemented strains compared with the wild-type 923. Panels **c**, **e**, pseudoiodinine production by different strains. The numbers on *y*-axes show pseudoiodinine yields. **c** The wild-type 923, *rsmY*, *rsmZ*, and *rsmYZ* mutants and their corresponding complemented strains in LB medium. Error bars show means ± SD ($n = 3$ biological independent replicates) and significant differences at ***$P < 0.001$, ***$P = 0.0002$, $= 2.5 \times 10^{-14}$, $= 1.5 \times 10^{-10}$ in sequence. **e** Pseudoiodinine yields in wild-type *P. mosselii* 923 in LB and TSB medium, the *csrA1*, *csrA2*, *csrA3*, *csrA1A2*, *csrA1A3*, *csrA2A3* and *csrA1A2A3* mutants, and overexpressing and RBS-swapped strains in TSB medium. Error bars show means ± SD ($n = 3$ biological independent replicates) and significant differences at ***$P < 0.001$, **$P < 0.01$, **$P = 0.005$,

***$P = 4.5 \times 10^{-10}$, $= 1.1 \times 10^{-9}$, $= 9.3 \times 10^{-17}$ in sequence. One-way ANOVA followed by LSD test. Experiments were performed three times independently with similar results. **f** Proposed model of GacA/CsrA-mediated regulation of pseudoiodinine biosynthesis in *P. mosselii* 923. In strain 923, the GacS/GacA TCS responds to uncharacterized signals, and GacA activates transcription of two non-coding RNAs (*rsmY* and *rsmZ*) by binding to their promoters, thereby positively regulating pseudoiodinine production. The two CsrA/RsmA homologs, CsrA2 and CsrA3, inhibit the initiation of pseudoiodinine synthesis. CsrA2 and CsrA3 interact predominantly with GGA-containing sequences/loops in the 5′-untranslated region of *psdA* mRNAs; this site overlaps the RBS and interferes with translation of the *psdA* transcript. RsmY and RsmZ contain multiple CsrA/RsmA binding sites within the predicted stem-loops that compete with mRNAs for CsrA2 and CsrA3 binding, thus antagonizing the suppression of translation.

The discovery of biocontrol microorganisms and the development of naturally occuring biopesticides is an effective way to control plant diseases and maintain global agricultural security. In this study, we report a potential BCA of *P. mosselii* 923 with specific antagonistic activity for *Xoo*, *Xoc*, and *M. oryzae*. Pseudoiodinine is the main antagonistic compound produced by *P. mosselii* 923, and the *psdABCDEFG* operon is sufficient for its production.

Biocontrol agents produced by *Pseudomonas* spp. have been used to control agricultural diseases, including phenazines, lipopeptide antibiotics, and other secondary metabolites[41]. Phenazine derivatives (e.g., PCA and pyocyanin) have been used for biocontrol of fungal diseases and have been studied in *P. fluorescens* 2–79, *P. chlororaphis* PCL1391 and *P. aeruginosa* 30–84[42]. *P. mosselii* is regarded as an opportunistic human pathogen distributed in the rhizosphere soil that is known to inhibit pathogens and insects[43,44]. For example, the *c-xtl* gene cluster involved in xantholysin biosynthesis was discovered in *P. mosselii* BS011 and inhibited the development of *M. oryzae*[45]. Recently, an engineered *P. mosselii* strain expressing the *ripAA* gene effectively

suppressed bacterial wilt in tobacco[46]. Moreover, a novel insecticidal protein from *P. mosselii* designated PIP-47Aa was effective in controlling the corn rootworm[47]. In the present study, we show that the wild-type *P. mosselii* 923 has the capacity to inhibit *Xoo* (PXO99A), *Xoc* (RS105), and *M. oryzae* (R01-1) (Fig. 1a, b). The antimicrobial compound pseudoiodinine produced by *P. mosselii* 923 can inhibit both *Xoo* (PXO99A) and *Xoc* (RS105) with $EC_{50}$ values of 0.17 μg/mL and 1.36 μg/mL, respectively (Table 1). Pseudoiodinine also shows strong fungicidal activity against *M. oryzae* with an $EC_{50}$ of 4.43 μg/mL (Supplementary Fig. 9c). To our knowledge, this is the first report describing *P. mosselii* 923 and pseudoiodinine as antimicrobials for both bacterial and fungal growth.

Forty years ago, the antimicrobial compound pseudoiodinine was identified as a product of *P. fluorescens* var. *pseudoiodinine*[22]. Pseudoiodinine is a member of the pyrazolo[4,3-e][1,2,4]triazine family of NPs[19]. Although chemical synthesis of pseudoiodinine has been reported[19], the yield is generally low and costs are high, thus limiting the feasibility of a synthetic source. Interestingly, the genetic basis of pseudoiodinine biosynthesis and assembly of the molecule has not been well-studied. In this study, GacA was shown to positively regulate pseudoiodinine biosynthesis (Fig. 3a and Supplementary Fig. 13a–c). Previous studies have shown that the conserved GacS/GacA TCS positively regulates the expression of multiple antimicrobial pathways in *Pseudomonas*, including gene clusters in *P. fluorescens*[48], *P. chlororaphis*[49] and *P. aureofaciens*[50]. In the current study, the ΔgacA mutant was impaired in pseudoiodinine production in *P. mosselii* 923 (Supplementary Fig. 13b, c), which led us to further investigate the biosynthesis and regulation of pseudoiodinine.

To identify the BGC involved in pseudoiodinine biosynthesis, we analyzed the *P. mosselii* genome and transcriptome in strain 923 and the ΔgacA mutant and found that genes encoding GTP cyclohydrolase and methyltransferase were significantly repressed in the ΔgacA mutant (Supplementary Table 2). Mutations were generated in a number of downregulated genes and screened for pseudoiodinine biosynthesis and antibacterial activity; this resulted in the identification of the *psdABCDEFG* operon (Fig. 3e). *psdA* and *psdF* encode methyltransferases, *psdB* encodes a member of the glyoxylase bleomycin resistance protein dioxygenase superfamily, *psdC* encodes a sulphatase-modifying factor, *psdD* encodes a GTP cyclohydrolase and *psdE* encodes a WD-40 repeat protein. Deletion of these six *psd* genes greatly abolished pseudoiodinine production, and complementation restored production (Fig. 3c, d), thus indicating that *psdABCDEF* is necessary for pseudoiodinine biosynthesis. The ΔpsdG mutant was only partially attenuated in antibacterial activity and pseudoiodinine production (Fig. 3b, c). Sequence analysis of *psdG* indicates that it encodes a RibD protein, which functions in riboflavin biosynthesis. It is well-established that riboflavin (vitamin B2) acts as a cofactor and signaling molecule[51], and *psdG* may have a role in the structural or functional stability of pseudoiodinine. Pseudoiodinine was produced in *P. putida* KT2440 when the *psdABCDEFG* operon was expressed in trans (Fig. 4a), which confirms that the *psdABCDEFG* operon is sufficient for pseudoiodinine biosynthesis in *Pseudomonas*.

The predicted amino acid sequences of PsdD and PsdG show significant similarity to genes encoding toxoflavin biosynthesis, which suggests that pseudoiodinine could be partially synthesized via a biosynthetic pathway shared with or similar to toxoflavin[33]. We propose that this reaction produces pseudoiodinine through a sequence of ring contraction and methylation steps (Fig. 4b). 1,6-DDMT is confirmed as a biosynthetic intermediate of pseudoiodinine when the mutant strains ΔpsdC or ΔpsdE was fed with it, and pseudoiodinine production was restored (Supplementary Fig. 18). However, 1,6-DDMT was not detected in either wild-type *P. mosselii* 923 or mutant strains, which we thought it quickly turn into the next intermediate. It should be pointed out that the formation of 1,6-DDMT mediated by PsdE and

PsdC remains unclear and a detailed ring contraction pathway to pseudoiodinine awaits further investigation.

Previous studies have demonstrated that GacA and CsrA/RsmA have opposing regulatory roles on target genes at the post-transcriptional level[52,53]. Our EMSA results demonstrated that GacA binds to the promoter regions of two noncoding small RNAs, *rsmY* and *rsmZ* (Supplementary Fig. 19b, c), and binding activated transcription of the pseudoiodinine synthetic gene cluster. Further bioinformatic studies indicated that *P. mosselii* 923 contained three homologous *csrA* genes, namely *csrA1*, *csrA2*, and *csrA3* (Supplementary Fig. 19d). Deletion mutagenesis and post-transcriptional *uidA* fusions were used to show that CsrA1, CsrA2, and CsrA3 negatively regulated *psdA* expression and production of pseudoiodinine (Fig. 5d, e and Supplementary Fig. 19e, f). Interestingly, a GGA site was present in the 5′ untranslated region of the *psdA* RBS, which suggested that CsrA could repress *psdA* expression by binding to the leader transcript and blocking ribosome access (Fig. 5f). Further experiments are needed to determine how CsrA interacts with target mRNA. However, the mechanism by which pseudoiodinine exerts its antimicrobial activity has yet been discovered, which warrants further study. This report further identifies GacA and CsrA as important regulators of pseudoiodinine biosynthesis, and broadens our understanding of pseudoiodinine as an antimicrobial for both bacteria and fungi.

Microorganisms are the source of many bioactive secondary metabolites including antibiotics, antitumor and antiviral compounds that can be used in agriculture and medicine[54]. However, wild-type strains isolated from nature generally produce limited amounts of secondary metabolites, which necessitates strategies for improving metabolite yield[55]. In the current study, we utilized a multi-pronged approach to improve pseudoiodinine yield including engineering an overexpressing strain, deleting the negative regulatory genes, modifying the RBS and optimizing the culture medium. These approaches led to development of a genetically modified strain that produced 22.4-fold more pseudoiodinine than the wild-type with average yields of 42.5 mg/L (Fig. 5e). Further methods to increase the productivity of pseudoiodinine are needed, including industrial scale fermentation. Other strategies to maximize product yields include promoter manipulation, genomics, protein engineering, metabolomics, and metabolite profiling[56,57], which have been used to increase PCA yields by *P. aeruginosa* strain PA1201[57].

Pesticides are widely used not only in China but also in the world for managing plant diseases. For example, Shenqinmycin (PCA) has been used to protect rice and vegetable crops from rice sheath blight, pepper blight and seedling damping-off[58]. Our study showed that pseudoiodinine can prevent BLB, BLS and rice blast caused by pathogens *Xoo*, *Xoc*, and *M. oryzae* without negatively affecting rice in the greenhouse (Supplementary Figs. 10–12); our field assays show that pseudoiodinine can significantly reduce lesion areas of BLB with a biocontrol efficiency of more than 50% at concentrations > 5 μM (Supplementary Fig. 3b, c).

In the process of analyzing other pseudomonads for the *psdABCDEFG* gene cluster, we found that it was conserved; however, there are no homologs of the *psd* operon in *P. fluorescens*. particularly, the type strain *P. mosselii* DSM17497 harbored a 6931 bp gene cluster (*orf2184*, *orf2185*, *orf2186*, *orf2187*, *orf2188*, *orf2189*, *orf2190*) with 99% amino acid similarity to the pseudoiodinine operon. Several differences were noted in the *P. mosselii* 923 and DSM17497 gene clusters as follows: (i) the amino acid at position 39 in Orf2185 is mutated from Lys (K) to Asn (N) in DSM17497 (Supplementary Fig. 16a); (ii) the residue at position 89 in Orf2186 is mutated from Thr (T) to Ala (A) (Supplementary Fig. 16b); and residues at positions 158 and 208 in Orf2190 are mutated from Tyr (Y) to His (H) and Ser (S) to Val (V) (Supplementary Fig. 16c), respectively.

Interestingly, *P. mosselii* DSM17497 did not produce detectable levels of pseudoiodinine and did not inhibit *Xoo* PXO99A. To investigate

whether the *P. mosselii* DSM17497 *psd* cluster was functional, we generated two replicate strains each of the *P. mosselii* 923 mutants Δ*psdB*, Δ*psdC*, and Δ*psdG* containing an introduced copy of the DSM17497 *psd* operon. Surprisingly, bacteriostatic activity and pseudoiodinine production were restored to the wild-type 923 level in the complemented strains (Supplementary Fig. 17a, b). Moreover, *orf2184-orf2190* were expressed in *P. mosselii* DSM17497 (Supplementary Fig. 17c), indicating that the pseudoiodinine gene cluster was transcribed in *P. mosselii* DSM17497. It remains possible that inhibitors or repressors may exist in DSM17497 that suppress the synthesis or export of pseudoiodinine or its precursors. The lack of pseudoiodinine production by *P. mosselii* DSM17497 warrants further investigation, which will hopefully improve the efficacy of *Pseudomonas* as a biocontrol agent.

In summary, our study describes *P. mosselii* 923 as a biocontrol agent and pseudoiodinine as a potential green biopesticide. The pseudoiodinine BGC and predicted biosynthetic pathway may encourage further studies on pathway engineering and large scale production. Furthermore, the underlying GacA-*rsmY*/*Z*-CsrA123 regulatory network can be deployed to engineer higher yields of pseudoiodinine (Fig. 5f). This is urgently needed since the diseases caused by *X. oryzae* and *M. oryzae* continue to limit yields of rice and require vigilant intervention methods.

## Methods

### Bacterial and fungal strains, plasmids, and growth conditions

Phytopathogenic *Xanthomonas* spp. were cultivated in nutrient agar (NA) at 28 °C[59], and *E. coli* strains DH5α and BL21(DE3) were cultured in Luria Bertani (LB)[60] medium at 37 °C. Tryptic soy broth (TSB; 30 g/L) was used for culturing *Pseudomonas* strains at 30 °C. The fungus *M. oryzae* R01-1 was grown in oatmeal (OAM)[61] medium at 25 ˚C. The strains, plasmids and primers used in this study are listed in Supplementary data 1 and 2. When required, antibiotics were added at the following final concentrations (µg mL$^{-1}$): kanamycin (Km), 25; rifampicin (Rif), 75; gentamicin (Gm), 20; and spectinomycin (Sp), 25. Chemical reagents including petroleum ether, ethyl acetate, n-butyl alcohol and methanol were purchased from Macklin (Shanghai, China). Phenazine-1-carboxylic acid (PCA) was provided by Dr. Yawen He (School of Life Sciences and Biotechnology, Shang Hai Jiao Tong University). Restriction enzymes were purchased from Takara Bio (Europe AB).

### Analytical methods

NMR spectra were recorded on a Bruker Avance III 600 MHz spectrometer (Germany). The samples were dissolved in deuterated methanol (CD$_3$OD) or deuterated chloroform (CDCl$_3$). ESI-MS analysis of compounds was carried out on an Agilent UPLC 1290 Infinity II/6545 Q-TOF (USA). Both semipreparative HPLC and analytical HPLC were conducted with an Agilent Technologies 1260 Infinity II HPLC with a DAD detector. Columns were as follow: Fisher Wharton C18 column (250 × 10 mm, 5 µm, USA) (for semipreparative HPLC) and Agilent C18 column (150 × 4.6 mm 5 µm, USA) (for analytical HPLC). 1,6-DDMT was purchased from Princeton Biomolecular Research (Catalog No. PBMR 232664).

### Antibacterial and antifungal tests

To isolate bacteria with antimicrobial activity for *Xoc* RS105, soil samples (*n* = 248) were collected from the rhizosphere[5]. Briefly, 10 g sample was mixed evenly with 90 mL sterilized water, then this suspension was diluted to different gradients with sterilized water. 100 µL above solution was plated onto NA medium containing RS105. Antagonism of soil bacteria towards *Xoc*, *Xoo* and other pathogens was assessed by the Kirby-Bauer (KB)[62] method. All strains were tested in triplicate and the inhibition zones towards pathogenic bacterial were measured 2–3 d after cultivation at 28 °C on NA and *M. oryzae* R01-1 were measured 5 d after cultivation at 25 °C on OAM.

## Genomic analysis and phylogenomics

The *16S rRNA* of strain 923 was amplified using universal primers 27F and 1492R (Supplementary Data 2) using established methods[5]. The complete gene was purified and sequenced by BioSune Biotech (Shanghai, China) and used for Blast searches in the National Center for Biotechnology Information (NCBI) database. The *16S rRNA* sequences for 14 representative *Pseudomonas* spp. were obtained from the NCBI database, and a phylogenetic tree based on *16S rRNA* was constructed using neighbor-joining analyses with MEGA 7.0[63]. A bootstrap test with 1000 replicates was used to evaluate the confidence level, and the internodes of branches represented percentage of confidence > 50%. The whole genome of 923 was sequenced using the Illumina Miseq PacBio sequencing platform at Personalbio (Shanghai, China). Average nucleotide identity (ANI) values were calculated using the J Species WS online service[64]. To determine the precise phylogenetic position of strain 923, all publicly available *Pseudomonas* genome sequences and related strains were retrieved from the NCBI GenBank database and included in the phylogenetic analyses according to the Type Genome Server (TYGS) (https://tygs.dsmz.de). The whole-genome sequence of strain 923 was deposited in the NCBI BioProject Database (BioProject ID: PRJNA826312).

## Isolation and purification of antibacterial metabolites from *P. mosselii* 923

Strain 923 was cultured in 50 mL LB broth at 30 °C with aeration at 220 rpm for 12 h, then diluted 1:100 into 2 L flasks containing 300 mL LB broth, and incubated for 24 h at 30 °C, 220 rpm. The fermentation was extracted three times with an equal volume of petroleum ether, ethyl acetate and *n*-butyl alcohol at room temperature, respectively. The organic and aqueous phases were concentrated with a rotary evaporator, freeze-dried, and resuspended in 1 mL methanol and sterile water, respectively. Fifty microliters of the above four samples were removed and assessed for antibacterial activity using *Xoo* PX099$^A$ as previously described[5].

The ethyl acetate extract exhibited the greatest antagonistic activity against *Xoo* PX099$^A$. A large-scale fermentation was carried out, and a 10 L batch of the fermentation was further extracted by ethyl acetate. The gelatinous crude extract was then loaded onto a C18 reversed-phase silica gel column (MeOH/H$_2$O, 3:7 to 9:1). Twelve fractions were obtained from the column and further purified by high performance liquid chromatography (HPLC) with a C18 column (Agilent C18 column, 250 × 20 mm, 5 µm, USA) (method A). Using this approach, seven compounds were isolated and designated PM-1, PM-2, PM-3, PM-4, PM-5, PM-6, and PM-7. The antagonistic activity of each compound was tested against PX099$^A$ after concentration with methanol as a control.

HPLC preparation and analysis method: Method A (preparation): The initial concentration of 15% methanol was maintained for 2 min and elution gradients proceeded as follows: linear gradient of methanol from 15 to 100%, 2–18 min; 100% methanol, 18–22 min; 100 to 15% methanol, 22–26 min; and 15% methanol, 26–30 min. The flow rate was 5 mL/min with a UV detector (wavelength: 210 and 500 nm). Method B (analysis): HPLC using solvents A (water) and B (methanol) with a flow rate of 0.4 mL/min (Agilent C$_{18}$ column, 150 × 4.6 mm 5 µm, USA, 210 and 500 nm). The initial concentration of 5% methanol was maintained for 4 min, followed by the following gradients: 5 to 20% methanol, 4–25 min; 20 to 100% methanol, 25–26 min; 100% methanol, 26–32 min, 100–5% methanol, 32–33 min, and 5% methanol, 33–40 min.

## Structural elucidation

The molecular formula of PM-3 was determined to be C$_6$H$_7$N$_5$O by UPLC/Q-TOF-MS (m/z 166.0722 [M + H]$^+$, calculated for C$_6$H$_8$N$_5$O, 166.0729) with six IHDs (index of hydrogen deficiency). The $^1$H and $^{13}$C NMR data (Supplementary Figs. 5, 6) showed the presence of

two methyl groups ($\delta_H$ 4.27, 4.44; $\delta_C$ 43.3, 57.7) and one aromatic proton ($\delta_H$ 9.14). Because there are too many quaternary carbon and nitrogen atoms in the molecule, it was difficult to further determine the planar structure of PM-3. Fortunately, a tiny crystal was generated in the methanol solvent by solvent evaporation after many attempts. Finally, the complete structure of PM-3 was determined by X-ray diffraction (Fig. 2c) and it was identical to pseudoiodinine.

## Quantitative analysis of pseudoiodinine production

Seed cultures (500 µL) were inoculated into 50 mL LB or TSB in 250 mL flasks. In order to extract and quantify pseudoiodinine, 1 mL of culture was collected and extracted twice with equal volumes of ethyl acetate. The organic phase was subsequently collected and dried at 30 °C. and the resultant residue was dissolved in 100 µL of methanol. Crude extracts (5 µL) were collected for HPLC analysis (Method B). Pseudoiodinine production was quantified using the peak area (A) in HPLC eluate according to the following formula: A = 776.86 pseudoiodinine (mM) + 52.234. The formula was derived from a calibration curve with purified pseudoiodinine, which had a correlation coefficiency ($R^2$) of 0.999. The experiment was repeated three times independently.

## Culture growth analysis

*P. mosselii* strains were cultured in 3 mL LB at 30 °C, 220 rpm for 12 h. Harvested cells were suspended to a final concentration of $OD_{600}$ = 2.0 using TSB medium. Serial dilutions were prepared and 100 µL aliquots were incubated on LB medium supplemented with the appropriate antibiotics at 30 °C overnight. For the growth curve, 10 mL of fermentation cultures in 50 mL flasks containing TSB were prepared as described above and incubated at 30 °C at 220 rpm. Growth of the cultures was followed by periodic measurement within 36 h by spectrophotometry at 600 nm ($OD_{600}$). This experiment was repeated three times independently.

## Biocontrol assays

Twenty seeds of rice cultivar Yuanfengzao were sown in plastic pots (16 × 12 cm), which were maintained in the greenhouse at 14 h light (30 °C)/ 10 h dark (28 °C) and 65% RH. Rice seedlings were used for biocontrol assays after 21 d of cultivation (3–6 leaf stage). Rice plants at the three-leaf stage are suitable for both *Xoo* and *Xoc* biocontrol assays. Rice plants at the six-leaf stage are suitable for *Xoo* and *Xoc* spray inoculations in the greenhouse. Cell suspensions of *Xoc* RS105 and *Xoo* PXO99$^A$ ($OD_{600}$ = 0.6) and *P. mosselii* 923 ($OD_{600}$ = 0.5) were prepared in advance. In brief, these strains were cultivated to mid-exponential phase, centrifuged, and washed twice with sterile distilled water to an $OD_{600}$ of 0.5–0.6. The supernatant of the 923 culture (SUP) was prepared by centrifugation at 4 °C, 13,800 × $g$ for 10 min. Pseudoiodinine stock solutions with MIC values of 4 µg/mL for *Xoc* RS105 and 0.5 µg/mL for PXO99$^A$ were prepared. Sterile distilled water (MOCK) was used as a control.

Biocontrol assays on rice seedlings at the three-leaf stage were inoculated by syringe injection. Briefly, Yuanfengzao rice leaves were inoculated with *Xoc* RS105-Gus ($OD_{600}$ = 0.6) at 3 h before (Tre) and after (Pre) inoculation with water, 923, Δ*psdA* ($OD_{600}$ = 0.5) or 923 SUP by needleless syringe, then maintained in the greenhouse. Disease lesion lengths were measured at 7 d. Yuanfengzao rice plants at the six-leaf stage were sprayed with bacterial cell suspensions (2 mL of $1.0 \times 10^8$ cfu mL$^{-1}$). Treatments were as follows: (1) rice plants inoculated with strains RS105 or PXO99$^A$; (2) rice inoculated with pathogens RS105 or PXO99$^A$ and then treated with water, strain 923 (2 mL suspension) or pseudoiodinine (5 mL) as Tre treatments; (3) rice inoculated with water, strain 923 (2 mL) or pseudoiodinine (5 mL) 12 h prior to spray-inoculation with RS105 or PXO99$^A$ as Pre treatments; and (4) sterile distilled water-inoculated controls (MOCK). 0.05% (v:v) Tween-20 (Sangon Biotech, Cat. No. A600560) was used in each treatments. Disease lesion areas were measured on leaves (n = 20) at 15 d and bacterial growth was measured at 7, 14, and 21 d after inoculation. Briefly, three, 4 cm pieces from leaf tips were excised with sterile surgical scissors, and treated with 75% ethanol (30 s), 3% sodium hypochlorite (3 min) and sterile-distilled water (1 min). Leaf material was then macerated with a tissue grinder (JX-FSTPRP) at 55 Hz (twice for 1 min) and then maintained at room temperature for 30 min. Serial dilutions were prepared and incubated on NA supplemented with the appropriate antibiotics at 28 °C for 3–4 d until single colonies could be counted (>100/plate). Above inoculation experiments were repeated three times independently.

For biocontrol assays in the field, rice cultivar Yuanfengzao was sown in paddies (1 × 2 m). In the field trial, only the flag leaves of rice plants were chosen for measurement in a five-point sampling method where one point includes 10 rice plants and one flag leaf/per plant. A total of 50 leaves were evaluated for each treatment. BLS and BLB disease severity was investigated according to the lesion area on leaves (n = 15) at 15 d after inoculation. Field experiments were repeated three times independently.

## *M. oryzae* infections analyses

Rice cv. CO39 (*Oryza sativa* L. subsp. *indica*) is highly susceptible to *M. oryzae* and was chosen for infection studies. Conidial suspensions were produced at a final concentration of $1 \times 10^5$ conidia ml$^{-1}$ by flooding 10-day-old *M. oryzae* culture plates with sterile distilled water containing 0.05% (v:v) Tween-20 as above. The suspension was then spray inoculated onto 2-week-old plants. Pseudoiodinine was sprayed onto the surface of rice leaves at a concentration of 40 µM at 24 h before (Pre) and after (Tre) inoculation with *M. oryzae* R01-1 spore. Plants were placed in storage boxes for 24 h to maintain high humidity with a 14:10 h light/dark photoperiod at 25 °C and 90% RH. Disease severity was recorded at 7 dpi by imaging. The relative area of lesions was calculated with Adobe Photoshop CS5. The experiment was repeated three times independently.

## β-glucuronidase (GUS) assays

Our previous research demonstrated that pHM1-derived vectors systems including pHG1 were efficient, stable and suitable for gene expression analysis in *X. oryzae* strains and for tracking infections in rice leaves[40]. Prior research with pHM1 vectors indicated that the intensity of GUS staining in rice leaves correlated with bacterial multiplication; thus, GUS activity of *Xoc* RS105-Gus-labeled strains in the rice tissues was assayed following our previous protocols[65]. The detailed method was described in the supplementary methods. Each treatment was replicated on three leaves, and three independent experiments were performed.

GUS activity in *P. mosselii* strains was evaluated using a modified protocol. Briefly, *P. mosselii* 923 and mutants were cultured in LB containing spectinomycin at 30 °C overnight. Bacterial cells were collected in 2 mL microcentrifuge tubes (three replicates), washed twice with 1 mL sterile water, and adjusted to an $OD_{600}$ of 0.5. Qualitative and quantitative GUS analyses of *P. mosselii* 923 and mutants were conducted as described in the supplementary methods.

## Construction of promoter-probe vectors

The multiple cloning sites (MCS) in the promoter probe vector pNG1 were positioned upstream of *uidA*, and the *Nde*I site overlapped with the translational start site (ATG). Thus we created *uidA* promoter-probe fusions at the transcriptional and post-transcriptional level. The *psdA* promoter region was cloned into pNG1 as an *Eco*RI/*Kpn*I fragment to create pNG1-P2064 with the *psdA* promoter-*uidA* fusion (Supplementary Data 1). Using a similar approach, the *psdA* promoter was cloned into pNG1 as an *Eco*RI/*Nde*I fragment, resulting in a translational

fusion with *uidA*; the resulting construct was designated pNG1-P2064-post (Supplementary Data 1).

### EZ-Tn5 mutagenesis screening and identification
Mutagenesis of *P. mosselii* strain 923 with EZ-Tn5 and the characterization of insertion sites was conducted using established protocols[5]. Briefly, 1 µL of the EZTn5 from EZ-Tn5™ <R6Kγori/KAN-2>Tnp Transposome™ Kit (Lucigen, TSM08KR) was electroporated into 923 competent cell, then screening mutants that absolutely losing or partially attenuated antagonistic activity against *Xoo* PXO99[A]. The rescued cloning of the EZ-Tn5 < R6Kγori/KAN-2> Transposon insertion site in the 923 genomic DNA was given according to the manufacturer's protocols.

### Generation of *gacA* deletion, complementation and over-expression strains
A *gacA* deletion mutant was generated by *sacB*-mediated double homologous recombination. Sequences flanking *gacA* at 716 bp upstream and 487 bp downstream were amplified by PCR using primers *gacA*-up-F1/*gacA*-up-R1 and *gacA*-down-F2/*gacA*-down-R2 (Supplementary Data 2), respectively. The upstream and downstream amplified products were gel-purified, digested with *Nde*I/*Kpn*I and *Xba*I/*Nde*I, respectively, and cloned into p2P24Km[66] to obtain construct p24-*gacA* (Supplementary Data 1). After ligation, this construct was introduced into *E. coli* DH5α competent cells and cultured on LB containing Km. Colonies containing p24-*gacA* were confirmed by PCR with M13-F and M13-R primers (Supplementary Data 2) and sequenced. Plasmid p24-*gacA* was introduced into *P. mosselii* 923 by electroporation, and colonies with kanamycin resistance and sucrose sensitivity were selected on NAN (NA without sucrose) containing Km (NAN + Km) and NAS (NA containing 10% sucrose) in succession. NAS-surviving colonies were individually cultured on NA and NA + Km, and colonies that grew on NA but not NA + Km were confirmed by PCR with primers in Supplementary Data 2. Deletion of *gacA* was confirmed by sequence analysis and further verified by testing for bacteriostasis and pseudoiodinine production as described above.

A 1074 bp fragment containing *gacA* and the upstream promoter region was amplified by PCR with primers *gacA*-com-F and *gacA*-com-R (Supplementary Data 2). The PCR product was cloned into pUFR034 as an *Eco*RI/*Kpn*I fragment to yield the recombinant plasmid pUFR-*gacA*; this construct was confirmed by PCR with M13 primers, sequenced, and then introduced into *P. mosselii* 923 and the Δ*gacA* mutant by electroporation. Colonies were selected on LB containing Km, identified by PCR and further confirmed by testing for bacteriostasis and pseudoiodinine production. The complemented mutant and *gacA* overexpressing strains were designated CΔ*gacA* and 923 pUFR-*gacA*, respectively.

### RNA-seq and transcriptome analyses
Wild-type *P. mosselii* 923 (WT) and the knockout mutant Δ*gacA* (KO) were cultured in LB for 24 h. Three biological replicates were prepared of each strain. Bacterial cells (1 mL) were collected and centrifuged for 2 min at 9600 × *g* at 4 °C; this was repeated with another 1 mL aliquot of cells. Supernatants were removed, frozen in liquid nitrogen for 15 min, and stored at −80 °C until needed.

RNA-seq was performed by Shanghai Personal Biotechnology Co., Ltd. with the Illumina HiSeq system. *DEseq* R package was used to analyze DEGs of KO compared to WT and a corrected *P*-value (q value) <0.005 and a log2 fold-change >1 were used to establish significance. Volcano plots were created using the R language *ggplots2* package and plot_volcano from soothsayer (https://github.com/jolespin/soothsayer) in Python v. 3.6.6. Heatmaps were produced by R language and the Pheatmap software package (https://rdrr.io/cran/pheatmap/). Euclidean and complete linkage methods were used to calculate distance and clustering, respectively.

### Quantitative RT-PCR
To verify RNA-seq data, three independent real-time quantitative PCR (RT-qPCR) analyses were performed on WT and KO samples using the same treatments as used for RNA-seq. Fifteen DEGs were randomly selected for secondary confirmation by RT-qPCR analysis. Total RNA of WT and KO samples were extracted using the EasyPure RNA Kit (Transgen Biotech). cDNA was prepared with the cDNA Synthesis Super Mix Kit (Transgen Biotech). SYBR green-labeled PCR fragments were amplified, and RT-qPCR was performed with an ABI7500 Real-Time PCR System (Applied Biosystems, USA). *rpoD* was used as an internal control and reference gene, and the $2^{-\Delta\Delta Ct}$ method was used for relative quantification[67]. All RT-qPCR reactions were performed three or more times independently using primers listed in Supplementary Data 2.

### Generation of *psd* deletion mutants and complementation analyses
Deletion of individual *psd* genes was accomplished by *sacB*-mediated double homologous recombination as described above. The primers used to amplify sequences flanking each gene are listed in Supplementary Data 2. Deletion mutants were identified by PCR with corresponding primers (Supplementary Data 2) and further evaluated for bacteriostasis and pseudoiodinine production. The other deletion mutants described in this study were also generated using *sacB*-mediated double homologous recombination.

For complementation analyses, *psdA, psdB, psdC, psdD, psdE, psdF*, and *psdG* and the *psdA* promoter were amplified and cloned in pBSPPc to obtain plasmids pBS-*psdA*, pBS-*psdB*, pBS-*psdC*, pBS-*psdD*, pBS-*psdE*, pBS-*psdF*, and pBS-*psdG*, respectively. (Supplementary Data 1). Constructs then were introduced into the corresponding deletion mutants and tested for complementation. The *rsmY* and *rsmZ* mutants were complemented through cloning in pUFR034 using the same protocol.

### Plasmid construction with One-Step ClonExpress technology
Plasmids pBSPPc-Pseu-ORF and pNG1-P2064 were constructed as described above. The primers used are listed in Supplementary Data 2 and were synthesized by Shanghai Generay Biotech Co., Ltd. Linearized pBSPPc-Pseu-ORF was obtained by reverse PCR with primers pBS-Pseu-F2 and pBS-Pseu-R2 and mixed with linearized pNG1-P2064 (Supplementary Fig. 20d); recombination proceeded using protocols supplied with the ClonExpress II One Step Cloning Kit (Vazyme, Nanjing, China). The recombination product, pBSPPc-P2064-RBS-Pseu-ORF, was used to transform recipient cells.

### Heterologous expression of pseudoiodinine in *P. putida* KT2440
The seven gene operon (*psdABCDEFG*) was cloned with its native promoter in pBSPPc. The 7.2-kb DNA fragment was amplified with primers pBS-Pseu-F and pBS-Pseu-R (Supplementary Data 2) and purified using a gel extraction kit. The 7.2-kb fragment was inserted into *Xba*I/*Bam*HI-digested pBSPPc and cloned using directions provided with the ClonExpressII One Step Cloning Kit. The resulting construct was designated pBSPPc-ABCDEFG; this was introduced into *P. putida* KT2440 through electroporation and verified by PCR.

The engineered *P. putida* strain was cultivated in 20 mL TSB supplemented with 20 µg/mL gentamicin, and cells were grown at 30 °C, 220 rpm, for 24 h. The fermentation broth was centrifuged, and the supernatant was extracted three times using equal volumes of ethyl acetate. The combined organic phases were concentrated and dissolved in methanol for HPLC analysis (Method B).

### Generation of *psd* operon overexpressing strains
For overexpression experiments, the plasmid pBSPPc-ABCDEFG (Supplementary Data 1) contained the *psdABCDEFG* operon and its

native promoter was introduced into *P. mosselii* Δ*csrA1A2A3* as described above and analyzed for pseudoiodinine production.

## Heterologous complementation
The pseudoiodinine homologous gene cluster (*orf2184-2190*) from *P. mosselii* DSM17497 and its native promoter was cloned as a *Xba*I/*Bam*HI fragment in pBSPPc, resulting in pBS-DSM17497-Pseu (Supplementary data 1). This clone was introduced into the *P. mosselii* 923 Δ*psdB*, Δ*psdC* and Δ*psdG* mutants to determine if the DSM17497 *psd* operon could complement the 923 mutant strains. Two strains of each mutant (*PsdB*-comp.1 and *PsdB*-comp.2; *PsdC*-comp.1 and *PsdC*-comp.2, and *PsdG*-comp.1 and *PsdG*-comp.2) were generated by introducing plasmid pBS-DSM17497-Pseu into the corresponding mutants by electroporation (Supplementary Data 1). Complemented mutants were evaluated for bacteriostasis and pseudoiodinine production as described above.

## RT−PCR analysis in *P. mosselii* DSM17497
DSM17497 contains seven genes homologous to the pseudoiodinine operon in *P. mosselii* 923, namely *orf2184* (*psdA*), *orf2185* (*psdB*), *orf2186* (*psdC*), *orf2187* (*psdD*), *orf2188* (*psdE*), *orf2189* (*psdF*) and *orf2190* (*psdG*). RT-PCR was used to determine whether the seven putative pseudoiodinine genes were expressed in *P. mosselii* DSM17497. Total RNA of *P. mosselii* DSM17497 was extracted using the EasyPure RNA Kit (Transgen Biotech), and cDNA was prepared with the cDNA Synthesis Super Mix Kit (Transgen Biotech). The *16S rRNA* gene was used for normalization. The primers used for RT-PCR are listed in Supplementary Data 2, and the experiment was repeated three times independently.

## Expression and purification of GacA
The *gacA* ORF was amplified by PCR with primers *gacA*-F and *gacA*-R (Supplementary Data 2) and cloned into pET28a as a *Nde*I/*Xho*I fragment. The resulting construct, pET28-*gacA*, was transformed into *E. coli* BL21 (DE3). For overexpression, a single colony *E. coli* BL21 colony was inoculated into 20 mL LB with Km and cultivated overnight at 37 °C. Cells (5 mL) were then transferred to 500 mL LB and grown at 37 °C, 220 rpm to $OD_{600} = 0.6$–0.8; cells were then induced with 0.5 mM IPTG and cultivated overnight at 16 °C. Pellets were harvested by centrifugation at 4 °C, $3,500 \times g$ for 10 min, washed twice in PBS, and suspended in buffer A (50 mM Tris-HCl, 300 mM NaCl, pH 7.5) containing a final concentration of 1 mM PMSF (phenylmethanesulfonyl fluoride). Cells were disrupted by sonication, and cell free extracts were obtained by centrifugation at 4 °C, $13,800 \times g$ for 30 min. Supernatants were applied to His Sep Ni-NTA Agarose Resin (Yeasen Biotechnology), which was equilibrated with buffer A prior to use. The Ni-NTA column was washed three times with buffer A, and the column was eluted with a gradient of 20–250 mM imidazole in buffer A. Fractions from 20 to 250 mM were pooled and separated by SDS-PAGE. The fraction containing GacA was concentrated and exchanged with storage buffer (50 mM Tris-HCl, 150 mM NaCl, 5% glycerol, pH 7.5). The protein concentration was estimated using a Nano-300 Micro-Spectrophotometer (Hangzhou Allsheng Instruments Co.), and the preparation was stored at −80 °C until needed.

## Electrophoretic mobility shift assays (EMSA)
Cy5-labeled promoter probes containing the UAS motif (TGTAAG-N6-CTTACA) in *rsmY* and *rsmZ* were synthesized commercially (Shanghai DNA Bioscience Co. Ltd). EMSA was performed using established protocols[68]. Briefly, the purified His-GacA was mixed with Cy5-labeled rsmY and rsmZ promoter fragments, respectively, then loaded on a 4.5% nondenaturing polyacrylamide gel for electrophoresis. The Cy5 fluorophore was detected using an Amersham Typhoon RGB Biomolecular Imager (Cytiva, Sweden). EMSA experiments were repeated three times independently.

## MIC and EC50 determination
The MICs of pseudoiodinine and PCA for *Xanthomonas* spp. were determined by serial dilution. NA was prepared containing pseudoiodinine at 0−64 µg/mL or PCA at 0−256 µg/mL. Two microliters of bacterial suspension ($OD_{600} = 1.0$) were diluted threefold and spotted to NA plates. After a 48 h incubation at 28 °C, MICs were defined as the lowest concentration at which no growth was visible.

$EC_{50}$ values of pseudoiodinine and PCA were determined according to growth inhibition. Briefly, 10 µL of bacterial suspension was added to 5 mL NB containing diluted concentrations of pseudoiodinine and PCA. $OD_{600}$ values of the tested suspensions were measured when the control suspensions increased to $OD_{600} = 1.0$. The log of percentage inhibition based on $OD_{600}$ values were regressed on the log of compound concentrations, and $EC_{50}$ values were calculated. This experiment was performed three or more times independently.

The $EC_{50}$ of *M. oryzae* R01-1 was determined in vitro by transferring plugs (0.5 cm² diameter) of mycelium from an actively growing fungal colony to a series of OAM plates containing pseudoiodinine at 10, 15, 20, 25, 30, 35, 40, 45 and 50 µM. Fungal colony diameters were measured after a five-day incubation at 25 °C in darkness, and inhibition was calculated as percent of the control growth. $EC_{50}$ values were calculated based on linear regression of colony diameter on log-transformed pseudoiodinine concentrations. Experiments were conducted three times independently.

## Statistical analyses and reproducibility
Statistical significance was calculated using the least significant difference (LSD) test method and one-way ANOVA analysis for multiple comparisons using SPSS v. 22.0. The statistical tests value represent statistically significant in the figure legends as follows: $*P < 0.05$; $**P < 0.01$; $***P < 0.001$, one-way ANOVA followed by LSD test. Data are presented as means ± standard deviation (SD). Statisticals data were analyzed using GraphPad Prism version 8.00. Lesions areas were calculated from infected leaves using Adobe Photoshop CS5. All experiments were repeated at least three times independently to confirm reproducibility.

## Reporting summary
Further information on research design is available in the Nature Portfolio Reporting Summary linked to this article.

## Data availability
The data that support the findings of this study are available within this manuscript and its Supplementary Information file. The genome sequence data of *Pseudomonas mosselii* 923 used in this study has been deposited in the NCBI GenBank database under the BioProject accession code PRJNA826312 (https://www.ncbi.nlm.nih.gov/nuccore/NZ_CP095556). The transcriptome data has been deposited to the NCBI Sequence Read Archive and is accessible with code PRJNA880759. Crystallographic data for the structure of pseudoiodinine reported in this Article has been deposited at the Cambridge Crystallographic Data Center, under deposition number CCDC 2175188. Copies of the data can be obtained free of charge via www.ccdc.cam.ac.uk. Strains, plasmids and primers used in this study are reported in the Supplementary Data 1–2. Source data are provided with this paper. Data is also available from the corresponding author upon request. Source data are provided with this paper.

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

## Acknowledgements

We are grateful to Prof. Carol Bender (Oklahoma State University), Prof. Feng Ge (Institute of Zoology, Chinese Academy of Sciences) and Prof. Xuewen Gao (Nanjing Agricultural University) for their critical view on this project. This work was supported by grants from the National Key Research and Development Program of China (2022YFD1400200, 2017YFD0200400), the National Natural Science Foundation of China (31830072) and the Shanghai Agriculture Applied Technology Development Program of China (2020-02-08-00-08-F01462).

## Author contributions

G.C. and L.Z. supervised the study. R.Y., Y.Y., She. L., Y.F., Y.L., Li. L., Lo. L. and Y.P. collected soil samples. R.Y. and She. L. isolated bacterial strains. R.Y. and L.Z. performed gene knockout, complementary, and biochemical assay experiments. R.Y. and Q.S. identified compounds. R.Y. and J.F. run biocontrol assays. Q.S., T.H. and X.W. performed biosynthesis experiments. R.Y. and Q.S. analyzed the data and wrote the manuscript. T.H., Shu. L. and G.C. revised the manuscript.

## Competing interests

The authors declare no competing interests.
