## [Peer Review File · Nature Communications]

The natural pyrazolotriazine pseudoiodinine from
Pseudomonas mosselii 923 inhibits plant bacterial and fungal
pathogensREVIEWER COMMENTS

Reviewer #1 (Remarks to the Author):

This is a terrific paper! The authors report a *Pseudomonas mosselii* strain (923) that makes a product that inhibits the growth of devastating rice pathogens. They demonstrate, through rigorous characterization, that the antimicrobial compound is pseudoiodinine, a previously known compound of the pyrazolo[4.3-e][1,2,4]triazine family, but with no known biosynthesis route. Here, the authors identify the biosynthetic gene cluster responsible for production of pseudoiodinine, the first biosynthetic proposal, and additionally, they use elegant genetic experiments to investigate the regulation of pseudoiodinine production. They thoroughly characterize the roles of a two-component signaling system, small regulatory RNAs and small RNA binding proteins. Using insights gleaned from the genetic analyses, the authors successfully engineer a *P. mosselii* overexpression strain that exhibits 22.4-fold higher yield of pseudoiodinine than the WT. In a model test, they show the promise of pseudoiodinine and/or *P. mosselii* strain 923 as a biocontrol agent. Indeed, the authors present compelling data that treatment with pseudoiodinine alone, cell free culture fluids containing pseudoiodinine, and inoculation with *P. mosselii* all provide striking protection against bacterial and fungal pathogens in a rice leaf model. What is also remarkable is that this protection occurs both a biocontrolled-greenhouse, as well as out in the field.

This work is especially thorough and impressively multi-disciplinary, combining natural product chemistry, bacterial genetics, and in planta experiments. Here are a few very minor suggestions for further consideration to enhance the significance of the work. Mostly I'm proving I read the paper as, in my opinion, it is so well done. It was a delight to read – crystal clear.

Experiments:

The only obvious experiment that was lacking is the following: The authors convincingly show that pseudoiodinine is sufficient to protect rice leaves from lesions. However, is it in fact the key and only compound contributing *in vivo*? The authors could evaluate a *P. mosselii* pseudoiodinine deficient mutant alongside the WT *in planta* to discover the necessity for pseudoiodine and/or if other antimicrobials contribute to plant protection. The authors have the mutant as they constructed it as part of this work (Figure 3).

Are there homologs of the *psd* operon in other known pseudoiodinine producers that could account for its production? (i.e., *P. fluorescens*)? It might be worth database search and a sentence or two.

The proposed biosynthetic model has the least experimental support. A detailed investigation is certainly beyond the scope of this paper. There could be a couple of simple experiments to bolster the proposed scheme: 1) can the authors monitor 1,6-DDMT in WT, $\Delta psdD$, and $\Delta psdG$ strains? or 2) can the authors feed GTP and/or glycine to the *P. putida* recombinant strain KT2240 and determine whether increased pseudoiodinine production occurs? Alternatively, proposed biosynthetic precursors/intermediates could be fed to the mutant(s) in the appropriate steps and the authors can monitor whether pseudoiodinine production is restored. I emphasize, this work is not required for publication in my view, but it enhances that aspect of the paper and I think it might be easy because the authors have the mutants the needed compounds.

Minor points:

It would be beneficial to provide the gene annotations for the pseudoiodinine biosynthetic pathway in Figure 3E rather than having to go back and forth between the text/supplement.

Please provide full page figures and/or a list of ¹H, ¹³C, HMBC, and HSQC NMR spectra, including all observed chemical shifts.

Only with those data is it possible for readers to evaluate the chemical analyses/identification of the pseudoiodinine compound.

In the methods, please provide fuller information about instrumentation used for NMR, HPLC, MS to

make it possible for readers to replicate the work if they choose.

In the methods, please provide more information about how pseudoiodinine production was evaluated, e.g. extraction method, culture scale, injection volume, quantitation, etc.

Line 199 – sentence cut off: “..the psuedoiodinine nonproducing strain.”

Line 201 – awkward phrasing: “..with the capacity to produce pseudoiodinine, showing that the seven genes are sufficient...”

Line 277-278 – incomplete sentence

Line 362- “more evidence.”

Reviewer #2 (Remarks to the Author):

The manuscript provided two great findings of a biocontrol agent (BCA), *Pseudomonas mosselii* 923. First, *P. mosselii* 923 is BCA to multi disease pathogens such as *Xanthomonas oryzae* pv. *oryzae* (Xoo), *X. oryzae* pv. *oryzicola* (Xoc), and *Magnaporthe oryzae* by producing pseudoiodinine, a natural product that belongs to heterocycle pyrazolo[4,3-e][1,2,4]triazine family. Second, the first time pseudoiodinine synthesis in microorganism is reported and the discovery of its regulation by GacA and CsrA in *P. mosselii* 923 is important to engineering pseudoiodinine overproduction strain. Please consider major and minor points below to improve the MS:

-Major points

1. Pseudoiodinine influence on fungal infection on rice plants should be performed.
2. The methods used were not convincing to clarify the improvement of Xoo and Xoc resistance. L489-L518, the infection method was not clearly described. What is the plant age for biocontrol assay with Xoo and Xoc for the needleless method and spray method, respectively? L491, three-leave-stage and six-leave-stage plants are greatly different. Besides, a clip inoculate method is usually used for Xoo treatment and infection by needle is usually used for Xoc treatment. As described in the field test, a spraying method was used. Authors should clarify the practicability of the method under field trial such as imaging the whole area with and without treatment. L517, 15 leaves were investigated after inoculation. I do not think 15 leaves are sufficient. Also, how were 15 leaves chosen from 50 plants, by severity? I supposed not all the leaves were infected by the spraying method.
3. Authors reported that there was no negative effect on rice (L382). This is important for the usage of *P. mosselii* as BCA in the future. Fig S6 is not sufficient to claim that. Plant growth and morphology of leaves should be recorded to prove this.
4. The introduction of biosynthetic operon and removing the CsrA binding sites greatly increased pseudoiodinine production. I wonder if pseudoiodinine overproduction strain improves antibacterial activity and causes a negative effect on *P. mosselii*.

-Minor points

1. Fig. 1A, bacterial and fungal growth tests should be grouped separately to distinguish CK labels.
2. Fig 1B, C, D, E there should be mock treatment for “Tre” and “Pre” separately, means leaves are infected with Xoo/Xoc then treated with water (Mock_Tre) and leaves are treated with water then infected with Xoo/Xoc (Mock_Pre).
3. L48, “...the cultivation of rice varieties with disease resistance (R) genes has been the most practical approach to controlling *X. oryzae*”. This statement is not true. A review article in 2019 (<http://dx.doi.org/10.17582/journal.pjar/2019/32.2.359.380>) summarized many important measures to control BLB.
4. L121, “twelve fractions were labeled A1-A12” should be added.
5. *M. oryzae* RO1-1 -> *M. oryzae* isolate RO1-1
6. Abbreviations are usually defined at the first use in the main text. Please check “CDS” in the Fig S2 legend, “TSB” in L270, and so on.

7. English revise: P1L17 pesticide leads, P2L45, incited is caused?, L51 resistance is chemical resistance? L33, which negatively regulates expression -> please clarify expression of which?, and so on.
8. Citation: L49, Ref #5 does not provide the information in the statement.
9. Fig 5. I suggest removing the rice plant clipart.

Reviewer #3 (Remarks to the Author):

In my opinion, the authors have undertaken a suitably thorough effort to characterise PM-3 (pseudoiodinine) using mass spectrometry (HRMS) and NMR (^1H , ^{13}C , HSQC, HMBC). Due to the simplicity of the spectra, it would likely not be sufficient to tell whether this structure was correct using these two techniques alone. However, the authors have also solved the crystal structure which appears to be consistent with their proposed molecular structure. As such, I am confident that the authors have characterised this compound correctly and no additional evidence is requested.

Point-by-point Responses (Re:) to the reviewer comments

Reviewer #1 (Remarks to the Author):

This is a terrific paper! The authors report a *Pseudomonas mosselii* strain (923) that makes a product that inhibits the growth of devastating rice pathogens. They demonstrate, through rigorous characterization, that the antimicrobial compound is pseudoiodinine, a previously known compound of the pyrazolo[4.3-e][1,2,4]triazine family, but with no known biosynthesis route. Here, the authors identify the biosynthetic gene cluster responsible for production of pseudoiodinine, the first biosynthetic proposal, and additionally, they use elegant genetic experiments to investigate the regulation of pseudoiodinine production. They thoroughly characterize the roles of a two-component signaling system, small regulatory RNAs and small RNA binding proteins. Using insights gleaned from the genetic analyses, the authors successfully engineer a *P. mosselii* overexpression strain that exhibits 22.4-fold higher yield of pseudoiodinine than the WT. In a model test, they show the promise of pseudoiodinine and/or *P. mosselii* strain 923 as a biocontrol agent. Indeed, the authors present compelling data that treatment with pseudoiodinine alone, cell free culture fluids containing pseudoiodinine, and inoculation with *P. mosselii* all provide striking protection against bacterial and fungal pathogens in a rice leaf model. What is also remarkable is that this protection occurs both in a biocontrolled-greenhouse, as well as out in the field.

This work is especially thorough and impressively multi-disciplinary, combining natural product chemistry, bacterial genetics, and in planta experiments. Here are a few very minor suggestions for further consideration to enhance the significance of the work. Mostly I'm proving I read the paper as, in my opinion, it is so well done. It was a delight to read; crystal clear.

Re: Thank you for the positive and professional comments. We were very excited to submit this paper to Nature Communications when we identified this pseudoiodinine.

Experiments:

The only obvious experiment that was lacking is the following: The authors convincingly show that pseudoiodinine is sufficient to protect rice leaves from lesions. However, is it in fact the key and only compound contributing in vivo? The authors could evaluate a *P. mosselii* pseudoiodinine deficient mutant alongside the WT in planta to discover the necessity for pseudoiodinine and/or if other antimicrobials contribute to plant protection. The authors have the mutant as they constructed it as part of this work (Figure 3).

Re: Thank you for this important point. As the reviewer suggested, we repeated our experiments to test the biocontrol activity of mutant $\Delta psdA$ (pseudoiodinine mutant) WT, and other strains of *P. mosselii* 923 on rice seedlings and plants. As shown in the

results below, the lesion lengths and the GUS activity of deletion mutant $\Delta psdA$ was not significantly different from the MOCK treatment (**Figure 1c, d, below**). Meanwhile, we tested the effects of mutant $\Delta psdA$, WT *P. mosselii* 923 and pseudoiodinine on growth inhibition of the pathogen *Xanthomonas oryzae* pv. *oryzae* (*Xoo*) in planta. As shown below, there was no significant difference in growth of *Xoo* when plants were inoculated with $\Delta psdA$ and the mock-inoculated control (new version of **Supplementary Figure S11**). These results indicate that pseudoiodinine functions in protecting rice from *Xoo*. (L152-L160)

Figure 1. Antagonistic and biocontrol activity of *P. mosselii* 923. c, *Xoc* RS105-Gus lesion lengths on Yuanfengzao rice inoculated with strain 923, $\Delta psdA$ or 923 supernatants (SUP) at 7 d post-inoculation (dpi) in the greenhouse. ‘Pre’ represents rice leaves pretreated with water (MOCK-Pre), 923, $\Delta psdA$ or 923 SUP prior to pathogen inoculation. ‘Tre’ represents rice leaves inoculated with *Xoc* RS105-Gus and then treated with water (MOCK-Tre), 923, $\Delta psdA$ or 923 SUP. d, Population dynamics of *Xoc* RS105-Gus in rice leaves. Bacterial populations were monitored by GUS quantification and histochemical staining at 7 dpi.

Supplementary Figure S11. Effects of mutant $\Delta psdA$, WT *P. mosselii* 923 and pseudoiodinine on growth inhibition of *Xoo* in *planta*. Rice leaves sprayed with strains $\Delta psdA/923$ ($OD_{600}=0.5$) or 0.5 $\mu\text{g/mL}$ pseudoiodinine (PSD) 12 h before (Pre) and after (Tre) inoculation with *Xoo* PXO99^A ($OD_{600}=0.6$) in the greenhouse. Lesion areas were quantified from infected leaves at 21 d post-infection ($n=20$ independent leaves). Statistical significance was assessed by the LSD test method with one-way ANOVA, ***, $P<0.001$.

Are there homologs of the *psd* operon in other known pseudoiodinine producers that could account for its production? (i.e., *P. fluorescens*)? It might be worth database search and a sentence or two.

Re: To our knowledge, besides *P. mosselii* strain 923, there is only one stain, *Pseudomonas fluorescens* var. *pseudoiodinum*, that has been reported as a pseudoiodinine producer (see Ref.22 *Chemische Berichte* 105, 1949-1955 (1972)). However, there are no genomic information has been reported. As a supplement, the homology analysis was carried out with other pseudomonads, we found that there are no homologs of the *psd* operon in *P. fluorescens* (L422-L423). Although *orf2184-orf2190* are existent and expressed in *P. mosselii* DSM17497 (Supplementary Fig. S17c), pseudoiodinine were not detected. The result has been discussed in the manuscript.

The proposed biosynthetic model has the least experimental support. A detailed investigation is certainly beyond the scope of this paper. There could be a couple of simple experiments to bolster the proposed scheme: 1) can the authors monitor 1,6-DDMT in WT, $\Delta psdD$, and $\Delta psdG$ strains? or 2) can the authors feed GTP and/or glycine to the *P. putida* recombinant strain KT2240 and determine whether increased pseudoiodinine production occurs? Alternatively, proposed biosynthetic precursors/intermediates could be fed to the mutant(s) in the appropriate steps and the authors can monitor whether pseudoiodinine production is restored. I emphasize, this work is not required for publication in my view, but it enhances that aspect of the paper and I think it might be easy because the authors have the mutants the needed compounds.

Re: Thank you for the comments on the proposed biosynthetic model of the pseudoiodinine, which we plan to investigate in the future. 1,6-DDMT was not detected in either wild-type *P. mosselii* 923 or mutant strains. We also fed GTP and glycine to *P. putida* KT2440; however, there was no significant change in pseudoiodinine production.

Most importantly, we also fed 1,6-DDMT to mutant strains $\Delta psdC$ and $\Delta psdE$ and showed that pseudoiodinine production was restored, indicating that 1,6-DDMT is indeed a biosynthetic intermediate. We added this result into the revised manuscript L241-L244 (Supplementary Figure S18).

Supplementary Figure S18. HPLC analysis (A_{500}) of pseudoiodinine production in mutant strains $\Delta psdC$ and $\Delta psdE$ that were fed with 1,6-DDMT.

Minor points:

It would be beneficial to provide the gene annotations for the pseudoiodinine biosynthetic pathway in Figure 3E rather than having to go back and forth between the text/supplement.

Re: Thanks for the comment. We added the gene annotations for the pseudoiodinine biosynthetic pathway into Figure 3e in the revised manuscript L1048-L1052.

Please provide full page figures and/or a list of 1H , ^{13}C , HMBC, and HSQC NMR spectra, including all observed chemical shifts. Only with those data is it possible for readers to evaluate the chemical analyses/identification of the pseudoiodinine compound.

Re: As suggested, we have provided the full page figures of 1H , ^{13}C , HMBC, and HSQC NMR spectra as Supplementary Figures S5-S8.

In the methods, please provide fuller information about instrumentation used for NMR, HPLC, MS to make it possible for readers to replicate the work if they choose.

Response: As suggested, we have added information about instrumentation used for NMR, HPLC, MS in the revised manuscript L464-L471.

In the methods, please provide more information about how pseudoiodinine production was evaluated, e.g. extraction method, culture scale, injection volume, quantitation, etc.

Response: As suggested, we have added information about quantitative analysis of pseudoiodinine production in the revised manuscript L529-L538. The pseudoiodinine inoculation volume was 5 mL as described in the revised manuscript (L568-L569).

Line 199 sentence cut off: the pseudoiodinine nonproducing strain;

Re: Corrected as requested. L213

Line 201 awkward phrasing: with the capacity to produce pseudoiodinine, showing that the seven genes are sufficient;

Re: Corrected as requested. L214-L215

Line 277-278 – incomplete sentence

Re: Corrected as requested. **L306-L307**

Line 362- “more evidence.”

Re: Corrected as further study. **L399**

Reviewer #2 (Remarks to the Author):

The manuscript provided two great findings of a biocontrol agent (BCA), *Pseudomonas mosselii* 923. First, *P. mosselii* 923 is BCA to multi disease pathogens such as *Xanthomonas oryzae* pv. *oryzae* (Xoo), *X. oryzae* pv. *oryzicola* (Xoc), and *Magnaporthe oryzae* by producing pseudoiodinine, a natural product that belongs to heterocycle pyrazolo[4,3-e][1,2,4]triazine family. Second, the first time pseudoiodinine synthesis in microorganism is reported and the discovery of its regulation by GacA and CsrA in *P. mosselii* 923 is important to engineering pseudoiodinine overproduction strain. Please consider major and minor points below to improve the MS:

Re: Thank you for the positive and helpful comments. We have tried our best to address these and believe that the revised manuscript is greatly improved.

-Major points

1. Pseudoiodinine influence on fungal infection on rice plants should be performed.

Re: As suggested, pseudoiodinine influence on fungal infection on rice plants have been added in this new submission (method of **L585-L595**). These results are included in Supporting Information as **Supplementary Figure S12**. As shown in the results, which suggested that pseudoiodinine also significantly reduced lesion area of rice blast caused by the fungus *M. oryzae* R01-1. **L165-L166**

Supplementary Figure S12. Effects of pseudoiodinine on rice leaf infection by *M. oryzae*. Pseudoiodinine (PSD) was sprayed onto the surface of rice leaves at a concentration of 40 μ M at 24 h before (Pre) and after (Tre) inoculation with *M. oryzae* R01-1 spore. The rice blast disease lesion area was calculated from $n = 20$ independently infected rice leaves 7 d post-infection. Statistical significance was assessed by the LSD test method with one-way ANOVA, ***, $P < 0.001$.

2. The methods used were not convincing to clarify the improvement of Xoo and Xoc resistance. L489-L518, the infection method was not clearly described. What is the plant age for biocontrol assay with Xoo and Xoc for the needleless method and spray method, respectively? L491, three-leave-stage and six-leave-stage plants are greatly different. Besides, a clip inoculate method is usually used for Xoo treatment and infection by needle is usually used for Xoc treatment. As described in the field test, a spraying method was used. Authors should clarify the practicability of the method under field trial such as imaging the whole area with and without treatment. L517, 15 leaves were investigated after inoculation. I do not think 15 leaves are sufficient. Also, how were 15 leaves chosen from 50 plants, by severity? I supposed not all the leaves were infected by the spraying method.

Re: Thank you for these comments. Rice plants at the three-leaf stage are suitable for both *Xoo* and *Xoc* biocontrol assays. Rice plants at the six-leaf stage are suitable for *Xoo* and *Xoc* spray inoculations in the greenhouse (displayed in L553-L555). Rice plants in the field are normally grown beginning in May until October when harvest occurs. We were unable to conduct a field trial this year (2022) in Shanghai because we received the comments on August 30th. Due to the Covid19 pandemic in China and repeated lockdowns, we were unable to conduct field trials elsewhere in China in fall 2022.

Field trials on pseudoiodinine and the pseudoiodinine-overproducing strains will be required prior to commercialization. In the interim, the biocontrol results in Shanghai rice fields are valid. We are convinced that the new data from greenhouse trials (**Supplementary Figure S11**) support the validity of our findings. In the field trial experiments, only the flag leaves of rice plants were chosen for evaluation in a five-point sampling method. One point includes 10 rice plants, and one flag leaf was measured from each plant. Thus, a total of 50 leaves were taken for measurement in each treatment. We have rewritten the corresponding methods and included them in the revised version (L579-L582). We apologize for the lack of clarity in the original version of the manuscript.

3. Authors reported that there was no negative effect on rice (L382). This is important for the usage of *P. mosselii* as BCA in the future. Fig S6 is not sufficient to claim that. Plant growth and morphology of leaves should be recorded to prove this.

Re: We were VERY cautious about potential deleterious effects of *P. mosselii* and pseudoiodinine on rice plant growth and leaf morphology. However, we observed no obvious changes to rice plants.

4. The introduction of biosynthetic operon and removing the CsrA binding sites greatly increased pseudoiodinine production. I wonder if pseudoiodinine overproduction strain

improves antibacterial activity and causes a negative effect on *P. mosselii*.

Re: Thank you for the very insightful thought. As suggested, we compared antibacterial activity and growth data of WT *P. mosselii* and the pseudoiodinine overproducing strain (methods of L540-L548). The pseudoiodinine overproducing strain improved antibacterial activity against the pathogens *Xoo* PXO99^A and *Xoc* RS105. However, there was a decrease in the growth of *P. mosselii* in the initial 14 h of fermentation (see the Figures below). We have included these observations in a new version of **Supplementary Figure S21**. (L315-L319)

Supplementary Figure S21. Antimicrobial and growth analysis. Antimicrobial activity of *P. mosselii* 923 and $\Delta csrA1A2A3$ -pBSPPc-P2064-RBS-Pseu-ORF strains for (a) *Xoo* PXO99^A and (b) *Xoc* RS105. c, The diameter of inhibition zones \pm s.d. (cm) ($n=3$). Growth of strains (d) *P. mosselii* 923 and (e) $\Delta csrA1A2A3$ -pBSPPc-P2064-RBS-Pseu-ORF on LB medium with 10⁻⁵ dilution of TSB. f, Growth of *P. mosselii* 923 and $\Delta csrA1A2A3$ -pBSPPc-P2064-RBS-Pseu-ORF in TSB medium. Samples were taken every 2 h for cell density (OD₆₀₀) measurements. Three independent biological experiments were performed with similar results.

-Minor points

1. Fig. 1A, bacterial and fungal growth tests should be grouped separately to distinguish CK labels.

Re: Suggested changes were made in the revised manuscript (Fig. 1a, b).

2. Fig 1B, C, D, E there should be mock treatment for 'Tre'; and 'Pre'; separately, means leaves are infected with Xoo/Xoc then treated with water (Mock_Tre) and leaves are treated with water then infected with Xoo/Xoc (Mock_Pre).

Re: We made these changes in the revised manuscript (Fig. 1c, d).

3. L48, 'the cultivation of rice varieties with disease resistance (R) genes has been the most practical approach to controlling *X. oryzae*'. This statement is not true. A review article in 2019 (<http://dx.doi.org/10.17582/journal.pjar/2019/32.2.359.380>) summarized many important measures to control BLB.

Re: We agree and we revised this sentence. The cultivation of rice varieties with disease resistance (R) genes appears to be better option to control *X. oryzae* than other management schemes (L49-L50).

4. L121, 'twelve fractions were labeled A1-A12'; should be added.

Re: Corrected as requested. (L123)

5. *M. oryzae* RO1-1 -> *M. oryzae* isolate RO1-1

Re: Yes. Normally, the species name and the strain name together indicate the isolate.

6. Abbreviations are usually defined at the first use in the main text. Please check 'CDS'; in the Fig S2 legend, 'TSB'; in L270, and so on.

Re: Thanks for pointing this out. Corrected as requested. CDS (coding sequences) in the Fig S2 legend; Tryptone soy broth (TSB) (L299), differentially expressed genes (DEGs) (L194).

7. English revise: P1L17 pesticide leads, P2L45, incited is caused?, L51 resistance is chemical resistance? L33, which negatively regulates expression -> please clarify expression of which?, and so on.

Re: Thank you for the suggestions. We fixed this. Incited means 'caused' (L47) and resistance is drug-resistance (L53), 'negatively regulate' refers to *psdA* (L36).

8. Citation: L49, Ref #5 does not provide the information in the statement.

Re: Thanks for pointing it out. We have revised this statement accurately according to the Ref 5 in the revised manuscript (see L51).

9. Fig 5. I suggest removing the rice plant clipart.

Re: We have removed as suggested.

Reviewer #3 (Remarks to the Author):

In my opinion, the authors have undertaken a suitably thorough effort to characterise PM-3 (pseudoiodinine) using mass spectrometry (HRMS) and NMR (¹H, ¹³C, HSQC, HMBC). Due to the simplicity of the spectra, it would likely not be sufficient to tell whether this structure was correct using these two techniques alone. However, the authors have also solved the crystal structure which appears to be consistent with their proposed molecular structure. As such, I am confident that the authors have characterised this compound correctly and no additional evidence is requested.

Re: Thanks for the positive comments. Yes, we were very lucky to obtain the crystal structure of pseudoiodinine.

REVIEWERS' COMMENTS

Reviewer #2 (Remarks to the Author):

I believe that the authors well addressed the comments raised by all the reviewers. Thus, the manuscript has been well revised. I have no additional comments. It is an excellent work!